# Isolation and Identification of *Fusarium* spp., the Causal Agents of Onion (*Allium cepa*) Basal Rot in Northeastern Israel

**DOI:** 10.3390/biology9040069

**Published:** 2020-04-02

**Authors:** Ben Kalman, Dekel Abraham, Shaul Graph, Rafael Perl-Treves, Yael Meller Harel, Ofir Degani

**Affiliations:** 1Plant Sciences Department, Migal–Galilee Research Institute, Tarshish 2, Kiryat Shmona 11016, Israel; kalman35@gmail.com (B.K.); dekel_abr@hotmail.com (D.A.); shaoulgraph@gmail.com (S.G.); 2The Mina and Everard Goodman Faculty of Life Sciences, Bar-Ilan University, Ramat Gan 52900, Israel; rafi.perl@gmail.com; 3Division for Plant Pests and Diseases, Plant Protection and Inspection Services, Ministry of Agriculture and Rural Development, Bet Dagan 50200, Israel; YaelM@moag.gov.il; 4Faculty of Sciences, Tel-Hai College, Upper Galilee, Tel-Hai 12210, Israel

**Keywords:** fungus, *Fusarium acutatum*, *Fusarium anthophilium*, *Fusarium oxysporum* f. sp. *cepae*, *Fusarium proliferatum*, molecular identification, morphological characteristics, pathogenicity assay, PCR detection, seed infection

## Abstract

Over the past decade, there have been accumulating reports from farmers and field extension personnel on the increasing incidence and spread of onion (*Allium cepa*) bulb basal rot in northern Israel. The disease is caused mainly by *Fusarium* species. Rotting onion bulbs were sampled from fields in the Golan Heights in northeastern Israel during the summers of 2017 and 2018. Tissue from the sampled onion bulbs was used for the isolation and identification of the infecting fungal species using colony and microscopic morphology characterization. Final confirmation of the pathogens was performed with PCR amplification and sequencing using fungi-specific and *Fusarium* species-specific primers. Four *Fusarium* spp. isolates were identified in onion bulbs samples collected from the contaminated field: *F. proliferatum*, *F. oxysporum* f. sp. *cepae*, and two species less familiar as causative agents of this disease, *F. acutatum* and *F. anthophilium*. Phylogenetic analysis revealed that these species subdivided into two populations, a northern group isolated from white (Riverside cv.) onion bulbs, and a southern group isolated from red (565/505 cv.) bulbs. Pathogenicity tests conducted with seedlings and bulbs under moist conditions proved that all species could cause the disease symptoms, but with different degrees of virulence. Inoculating seeds with spore suspensions of the four species, *in vitro*, significantly reduced seedlings’ germination rate, hypocotyl elongation, and fresh biomass. Mature onion bulbs infected with the fungal isolates produced typical rot symptoms 14 days post-inoculation, and the fungus from each infected bulb was re-isolated and identified to satisfy Koch’s postulates. The onion bulb assay also reflected the degree of sensitivity of different onion cultivars to the disease. This work is the first confirmed report of the direct and primary cause of *Fusarium* onion basal rot disease in northeastern Israel. These findings are a necessary step towards uncovering the mycoflora of the diseased onion plants and developing a preventive program that would reduce the disease damage.

## 1. Introduction

Onion (*Allium cepa* L., a member of the Amaryllidaceae family, also known as the bulb onion or common onion) is one of the most commonly grown and consumed vegetable crops in the world and is an important crop in Israel, grown during the winter–spring and harvested during the hot summer. In 2013, onion bulb production in Israel covered an area of ca 2500 ha throughout the country [1]. In 2017, the commercial production of onion bulbs for the local markets reached ca. 47,000 tons (9% of the bulk fresh vegetables according to data from the Israel Ministry of Agriculture and Rural Development). Species of the fungus genus *Fusarium* are widely distributed in soils in all climatic zones around the globe, associate with a vast diversity of plants, and cause severe plant diseases in crops [2]. Some members of this genus are responsible for diseases that have a significant economic impact in several onion-growing countries [3]. Wilting of onion plants with rot in the roots and basal plate of the bulb has been related mostly to *Fusarium oxysporum* f. sp. *cepae* [4], but formae speciales (f.sp., a group of isolates that are pathogenic to particular host plants) of *Fusarium solani* are also common in the roots of onions and cause plant wilting [5]. In addition, *Fusarium proliferatum* was more recently reported as the predominant fungal species isolated from the roots and bulbs of onion and garlic plants [6].

Pathogenic *F. oxysporum* is defined as one of the world’s most damaging plant pathogens [7]. It is a common soil-borne pathogen that has an exceptional level of host specificity with over 120 different formae speciales [8]. *F. oxysporum* f. sp. *cepae* causes severe disease in onions, with recorded yield losses of more than 50% [4]. Chlamydospores, produced by the older mycelium, enable *F. oxysporum* f. sp. *cepae* to survive in the absence of its host in the soil for a very long time. *Fusarium proliferatum* is related to *Fusarium oxysporum* f. sp. *cepae* and *Fusarium culmorum* but can be distinguished from these species based on morphology, and, to some extent, symptoms on the hosts. *F. proliferatum* can survive for extended periods in soil, so it is best to avoid planting the same field repeatedly with either onion or garlic [9]. *F. proliferatum* was reported in onion in different countries, including the United States [10], Serbia [6], Argentina [11], The Netherlands, Uruguay [12], Sweden [13], and Germany [14].

The accumulated knowledge of farmers in northern Israel and the Extension Service indicates that, in infested soil, there is an increasing risk of reduced seed germination and the appearance of onion basal rot throughout the growing season until harvest. According to these cumulative data, high temperatures before and during bulb maturation are optimal for fungus infiltration, establishment, and spread in the field. The onset of the disease usually appears as individual diseased plants or diseased patches in the field, and not in the entire field. Over time, the spread of the disease increases, more patches appear in the field, and the entire field can be affected. Crop protection methods are currently based on a four-year crop cycle and general soil disinfection with metam sodium before sowing.

Onion basal rot disease has been recognized in Israel for many years, causing about 1% yield loss in the fields (data from the Israel Ministry of Agriculture and Rural Development, and the Extension Service). However, the effect on yield and bulb quality has led to reduced growing areas and economic losses. Infected onions do not always show external disease symptoms, and if they reach storage, the problem is considerably worsened. When stored in open sheds or packing houses, the disease can soon spread to the rest of the onions. Moreover, there is concern that infected bulbs without visual symptoms will reach markets across the country. This concern is heightened by the presence of known toxins secreted by these pathogens [6]. Significant gaps of knowledge currently exist in Israel regarding the nature and distribution of the disease, the causal pathogens, and effective measures of control. Specifically, no disease-resistant onion cultivars have been recognized, and no fungicides have been proven to be unequivocally efficient and useful against the pathogen.

The growing number of reports in the past decade from local Israeli growers and consultation service personnel on onion root and bulb rot is a significant concern. Abraham Gamliel et al. (Southern Arava R&D annual report) reported in 2013 on an *F. proliferatum* disease outburst in white onions from different fields in the southern Arava area (southern Israel), probably caused by planting infected bulbs. Similar phenomena were reported in lily (*Lilium longiflorum*) and *F. proliferatum* (white and purple isolates), as well as *Fusarium oxysporum* were identified [1] as the causative agents. Subsequent pathogenicity trials conducted on lily bulbs and onion seedlings in growth chambers fulfilled Koch’s postulates and proved that the most pathogenic fungus was *F. proliferatum* (white isolates). Still, the disease in the Golan Heights in northeastern Israel has not been studied, and no data exist on the mycoflora of infected onion plants in this area. Thus, the objectives of the present study are to identify and characterize the causal agents of onion root and bulb rot in plants grown in commercial fields in this region of the country.

## 2. Materials and Methods

### 2.1. Isolation and Identification of Pathogens from Diseased Onion Plants

The study focused on two common varieties, Riverside (Orlando) cv. (white onion) and 565/505 cv. (red onion) (supplied by Hazera Seeds Ltd., Berurim MP Shikmim, Israel). Onion bulbs from commercial fields located in the Golan Heights (northeastern Israel) were analyzed for infestation with *Fusarium* species. Random samples of diseased onion plant parts, including onion roots and bulbs, were collected from two commercial fields at the end of the 2017 summer growing season (Kibbutz Ortal, northern Golan Heights) and the 2018 spring season (Moshav Eliad, southern Golan Heights). Bulbs were washed thoroughly under tap water, and the infected tissue of the roots and basal plate was separated using a scalpel. The tissue was cut into small segments (approximately 5 mm long), and each piece was surface-sterilized by dipping it in 70% alcohol for a few seconds, followed by immersion in 1% sodium hypochlorite for 1 min and then washing twice in sterile double-distilled water (DDW). After drying the tissue on a sterile paper towel in a fume hood, the bulb basal plate segments were plated on potato dextrose agar (PDA; Difco Laboratories, Detroit, MI, USA) medium supplemented with 2.5 mg/mL chloramphenicol. After incubation of the Petri dishes for 2–3 days at 28 ± 1 °C in the dark, the fungi started to grow from the tissue and were separated and transferred to a new plate. In order to obtain a pure colony, the single-spore subcultures method was used.

Cultures were propagated from single conidia, as follows. A 6-mm-diameter culture disc was transferred from each of the inspected isolates to a 15-mL test tube with DDW. After thorough vortexing, seeding of the fungus was performed by diluting the suspension up to a concentration of 1–5 spores per 5 µl. Spore counting using a Fuchs-Rosenthal counting chamber was done with a light microscope equipped with a Moticam 5 camera (Motic Instruments, Richmond, Canada). Then, 5 µl of each suspension was transferred to PDA plates and dispersed using a Drigelski stick. After two days of incubation in the dark at 28 ± 1 °C, the fungus grew on the new PDA plates. These colonies served as a single pure genetic source (common genetic origin) for further research. In this way, a total of 13 isolates was obtained. The isolates were divided into subgroups with identical colony characteristics, and representative fungal isolates from each group were selected and classified using taxonomic keys, including colony structures and macroscopic characterization. *Fusarium* spp. isolates were also molecularly identified as detailed below. In order to assess each isolate colony growth rate, radial mycelial growth was taken at two-day intervals (2, 4, 6, and 8 days) after inoculation by measuring the diameter along two perpendicular lines from the underside of the Petri dishes.

### 2.2. Microscopy

For microscopic observation of hyphae and spores formation, all isolates were grown on PDA plates for four days. Mycelial mats or conidia were scraped off the plate, and a small amount was suspended in 10 µl potato dextrose broth (PDB) or DDW and placed on sterile glass slides for microscopy observation. In order to maintain the integrity of the fragile fungal reproductive structures, the fungal slide culture technique [15] was used. A block of inoculated nutrient agar (taken from 2–4-day-old PDA colony) sandwiched between two sterile cover glass slides was placed in a plastic Petri dish containing water agar. The dishes, covered and sealed with parafilm, were incubated for 3–4 days at 28 ± 1 °C in the dark. The new hyphae and fungal reproductive structures were observed and photographed using a light microscope equipped with a Moticam 5 camera. The microscope magnification was 250:1, and no stain was used.

### 2.3. Isolation and Purification of Fungal DNA and PCR Amplification

Mycelia from PDA-grown colonies were used for DNA extraction using the Master Pure Yeast DNA Purification Set Kit (Sigma, Rehovot, Israel). Molecular identification by PCR and sequencing was made by targeting the *Fusarium* translation elongation factor-1 alpha (TEF1) gene and the fungus T12 beta-tubulin gene, as well as the small subunit ribosomal RNA gene, universal internal transcribed spacer (ITS). The *Fusarium oxysporum* f. sp. *cepae* (FOC) secreted in xylem genes 3 (SIX3, [16]) was used to determine whether the *F. oxysporum* samples belong to formae speciales *cepae*. The partial calmodulin gene sequence (CLPRO, [17]) was used to identify the *Fusarium proliferatum* isolate. Final confirmation of similarities between the isolates was achieved by the inter simple sequence repeat (ISSR)–PCR molecular method. The ISSR–PCR molecular technique provides a quick, reliable, and highly informative system for DNA fingerprinting using microsatellite sequences as PCR primers (Table 1), to generate multilocus markers [18]. DNA markers were amplified using the primer sets described in Table 1. The specific identification of *F. oxysporum* f. sp *cepae* was made using new primers designed in this work. The FOC–SIX3 was designed using Primer3Plus software (http://primer3plus.com) on a region of the SIX3 coding sequence common to *F. oxysporum* f. sp *cepae* and *Fusarium oxysporum* f. sp. *lycopersici*.

PCR was done using the Rapidcycler (Idaho Technology, Salt Lake City, Utah 84108, USA) in a total volume of 20 μL per reaction: 1 μL of each primer (concentration of 20 μM), 10 μL of commercial reaction mixture: RedTaq^®^ ReadyMix (Sigma, Rehovot, Israel), 3 μL of template DNA and 5 µl autoclaved DDW. PCR conditions: 94 °C for 2 min, 30 rounds of 94 °C for 30 s, 55 °C for 30 s, 72 °C for one minute, and a finishing step of 72 °C for 5 min [19]. For the FOC–SIX3 amplification, PCR reactions were performed using the following conditions: denaturation at 95 °C for 5 min; 35 cycles of denaturation at 95 °C for 15 sec, annealing at 61 °C for 15 sec, extension at 72 °C for 15 sec; final extension at 72 °C for 10 min, followed by cooling at 12 °C until recovery of the samples. For CLPRO amplification, PCR reactions were performed using the following conditions: denaturation at 95 °C for 5 min; 35 cycles of denaturation at 95 °C for 50 sec, annealing at 56 °C for 50 sec, extension at 72 °C for 1 min; final extension at 72 °C for 7 min, followed by cooling at 4 °C until recovery of the samples. For size determination and gel purification prior to sequencing, the PCR products were separated on 1.5% agarose gel electrophoresis (Lonza, Rockland, USA), and afterward sequenced (Hy Labs, Rehovot, Israel).

### 2.4. Identification of the Fusarium Species and Phylogenetic Relationships

Sequences were used to conduct a homology search against the GeneBank using the BLASTN tool [24] (nucleotide blast on the NCBI website, https://blast.ncbi.nlm.nih.gov/Blast.cgi, Table 2). Sequences were aligned using the Clone Manager 10.0 program (Sci Ed Software, Durham, NC, USA). The alignment of the TEF1 gene was used to construct a phylogenetic tree. The SeaView version 5.0 program (http://doua.prabi.fr/software/seaview) was used to generate a phylogenetic tree. The phylogenetic tree was performed using bootstrap with 1000 replicates and excluded positions with gaps.

### 2.5. Seedling Pathogenicity Assay

The pathogenicity test on onion seedlings was designed to measure the level of virulence of the *Fusarium* spp. isolates. The test was repeated twice with six replicates (Petri dishes with 10-15 seeds each). The Riverside (Orlando) cv. (white onion) and Noam cv. (red onion, supplied by Hazera Seeds Ltd., Berurim MP Shikmim, Israel) seeds were rinsed in DDW, soaked in NaOCl 1% for 1 min, and rinsed again twice in DDW. Each group of onion seeds was transferred to a Petri dish in which sterile Whatman paper was soaked in water. The seeds were inoculated with a 6-mm-diameter disc cut from a 5-day *Fusarium* sp. colony, grown previously on PDA, in the dark, at 28 ± 1 °C. A sterile 6-mm-diameter PDA disc was added to the control group. Sterile tap water was added to each plate every three days to maintain moisture and to allow efficient germination and disease progression. After nine days of incubation in the dark, at 28 ± 1 °C, the seeds were photographed, washed, and their germination percentages, sprout biomass, and hypocotyl length were measured. A germinating seed was defined as a seed in which the seed coat was broken by the radicle.

### 2.6. Onion Bulb Pathogenicity Assay

For the Koch’s postulates accomplishment, an onion bulb pathogenicity assay was conducted on Riverside (Orlando) cv. and Noam cv. (white and red onion cultivars, respectively). The entire assay was performed in six repetitions per isolate in a sterile environment within a biological hood. From each isolate, a stock of 10^6^ spores/mL (in sterile water) was prepared from a five-day-old colony grown previously on a PDA at 28 ± 1 °C in the dark. After removing the outer scales, the bulbs were sterilized in 70% ethanol and dried. Inoculation was done by stabbing the basal plate with a sterile pipette tip (10 mm in diameter) five times in different areas at a depth of 5 mm. Fifty µl spores were pipetted into each puncture, while a similar volume of sterile water was injected into the control bulbs. Each bulb was kept individually in a closed sterilized plastic bag to maintain a moist environment and prevent unwanted contamination, in a temperature-controlled incubator in the dark at 28 ± 1 °C. After two weeks of incubation, external and internal disease symptoms were photographed and evaluated qualitatively, and the fungus from each infected onion was re-isolated on PDA and identified to satisfy Koch’s postulates.

### 2.7. Statistical Analysis

A fully randomized statistical design was used in all experiments. Statistics and data analysis were carried out using the JMP program, 7th Edition, SAS Institute Inc., Cary, NC, USA. Disease symptoms were analyzed by one-way ANOVA followed by post-hoc multiple comparisons of Student’s *t*-test for each pair. The *t*-test was conducted with a significance threshold of *p* = 0.05 to compare the treatment means to those of the control.

## 3. Results

### 3.1. Isolation of Pathogens from Diseased Onion Plants

The disease symptoms observed on onions plants sampled in this study were dehydration of the flowers (Figure 1) and bulb rot spreading from the onion basal plate upwards in the scales, resulting in discolored and watery bulb tissue (Figure 2). We isolated single-spore cultures from these field-collected fungi and obtained six different isolates from diseased white and red onion bulbs. These bulbs were collected randomly from two commercial fields in the Golan Heights in northeastern Israel.

### 3.2. Molecular Identification of the Fusarium Isolates

Confirmation of the pathogen species was performed by PCR amplification and sequencing using the universal fungi and plants ITS, the fungous specific T12 beta-tubulin gene, and the *Fusarium* species-specific TEF1 primers. The amplified fragments (Appendix A) had the expected size (Table 1, Figure 3A). Amplicons were sent for sequencing, and the sequences were aligned against the GeneBank (NCBI, nucleotide blast). All four species examined presented high homology (99.4%–100% similarity) to previously described *Fusarium* spp. sequences in the GeneBank (Table 2). The identification of the *F. oxysporum* isolates as f. sp. *cepae* relies on the highest sequence similarity scores of the studied genomic regions (Table 2, Figure 3A), and on the amplification of a product at the expected length when using the new specific FOC-SIX3 primers designed in this work (Table 1, Figure 3B1). Additional strength and support to the identity of the *Fusarium* isolates achieved by the *Fusarium proliferatum* specific CLPRO gene identification (Figure 3B2). To identify similarities between the isolates, the inter simple sequence repeat (ISSR)–PCR molecular method was used. Indeed, the isolates from the same species, *F. acutatum* (B5 and B7) and *F. oxysporum* f. sp. *cepae* (B8 and B14), had very similar DNA fingerprinting (Figure 3C).

### 3.3. Morphological and Microscopic Characterization and Identification of the Fusarium Isolates

#### 3.3.1. Morphology of Plated Cultures

Six isolates recovered from rotting onion bulb plants were characterized in detail and assigned to four species based on molecular data (Table 2) and microscopy (see below). The isolates grew well on solid and liquid-rich media. On PDA solid media, the colonies usually reached 90 mm in diameter within 5 to 9 days at 28 °C under dark conditions. The colony growth rate differed among isolates (Table 3). *F. proliferatum* (isolate B1) was the fastest-growing among the isolates evaluated in this study, reaching a growth rate of 1.40 cm/day, *F. acutatum* (B5, B7) had an average growth rate of 1.07–1.24 cm/day, while the other two species, *F. oxysporum* f. sp. *cepae* (B8, B14) and *F. anthophilium* (B16), grew more slowly (0.83–0.87 cm/day). The colonies had different colors; their morphology varied among the species, and also within the same species (Figure 4). Cultures also showed age-dependent changes. The culture surface was flat and white at the beginning, and as the colony matured, typical colors for each *Fusarium sp.* appeared, usually purple, white, and gray (and sometimes light brown). When the cultures grew old, they became thick due to the growth of white aerial hyphae on the colony surface.

#### 3.3.2. Microscopy

Overall, the microscopical analysis confirmed that all cultures belonged to the genus *Fusarium*. Hyphae were 3.1–3.6 µm in width (Table 3), hyaline, septate, and branched (Figure 5A). Sporulation occurred within a few days after transferring the colonies to new plates. The *Fusarium* species produced three types of vegetative spores: microconidia, macroconidia, and chlamydospores, which varied in proportion among the species (Table 3). The most commonly produced spores were microconidia (Figure 5B, small cells) that were typically devoid of septa, and their shape varied from oval to kidney-shaped. Macroconidia (Figure 5B, large cells) had a characteristic falcate shape making them easily identifiable. In addition, they typically had three or four septa. Chlamydospores produced in or on older mycelium had one or two round cells and thick cell walls, which defend the cells against degradation and antagonists. Conidiophores (Figure 5C,D) were also hyaline and sometimes branched, and carried large quantities of microconidia. Conidia germinated rapidly, usually via monopolar or bipolar germ tubes (Figure 5B).

### 3.4. Phylogenetic Relationship

Sequences alignment of the TEF1 gene sequences from the six *Fusarium* isolates revealed a highly conserved 315 nucleotide array (Figure 6B). Phylogenetic analysis of those sequences supported the BLAST; nucleotide similarity search results are presented in Table 2. The isolates belonging to the same species, *F. acutatum* (B5 and B7) and *F. oxysporum* f. sp. *cepae* (B8 and B14), share the same branch (Figure 6A). Most interesting, the *Fusarium* isolates were grouped into two subclades. These subclades were according to the origin of the isolates. Three isolates (B1, B5, and B7) belong to a clade of isolates originating from the northern Golan Heights, while the second clade gathers the southern Golan Heights isolates (B8, B14, and B16). Moreover, the two groups of the *Fusarium* species were also isolated from different onion cultivars (white and red); thus, the onion cultivar may be the cause of the two unique mycoflora.

### 3.5. Seedling Pathogenicity Assay

*Fusarium* spp. can infect onion plants in many different ways, and the visual symptoms can be observed on plant leaves, roots, basal stem plate, and the bulb scales of small seedlings, mature plants, and dormant bulbs [25,26]. The seedling pathogenicity assay and the onion bulb inoculation assay (that will be detailed in the following section) are common investigation methods in plant pathology. These assays are widely used in various cases of host–pathogen interactions with some adaptions and variations, as was done in this study. The onion seedling assay was adjusted as an effective and rapid way to inspect the virulence capability of the *Fusarium* isolates. After nine days of incubation, it can be seen from Figure 7 that the *Fusarium* white mycelia grew on or near the onion seeds, apparently exploiting them as a food source.

The presence of *F. proliferatum* (isolate B1) or *F. acutatum* (B7, B5) significantly suppressed, and the presence of *F. oxysporum* f. sp. *cepae* (B8/B14) or *F. anthophilium* (B16) significantly increased, the seed germination of onion seeds (Riverside cultivars), under controlled conditions in Petri dishes after nine days compared to the control (*p* < 0.05, Figure 8). No difference in the damping-off effect was found between *F. oxysporum* f. sp. *cepae* (B8/B14) or *F. anthophilium* (B16) and the control. The Noam cv. seeds’ germination was not affected by the inoculation with the *Fusarium* isolates B1, B5, B7, and B14. However, germination was significantly enhanced by isolates B8 and B16 (*p* < 0.05). Nevertheless, inoculation of all six isolates had a significant adverse effect on growth, expressed as fresh biomass and hypocotyl elongation (difference from the control of *p* < 0.05 or more). Isolates B1, B5, and B16 were the most virulent according to these two parameters.

### 3.6. Onion bulb Inoculation Assay

Intact onion bulb virulence assay is another fast way to study the ability of the *Fusarium* isolates to colonize and develop in the host tissues. Inoculating onion Riverside cv. and Noam cv. bulbs with each of the *Fusarium* isolates led to symptoms development one week after inoculation. The pictures in Figure 9 were taken two weeks post-inoculation. Infection was apparent from the development of bulb tissue decay, accompanied by the outgrowth of white hyphae on the bulb surface. The symptoms that developed in the bulb inoculation assay were similar to those seen in naturally infected onion fields. Based on the qualitative estimation presented here, the onion bulb assay may reveal the degree of sensitivity of the two onion cultivars to the disease. Specifically, according to this test in the *F. oxysporum* f. sp. *cepae* (B8/B14) injection, Noam cv. had less severe symptoms than Riverside cv.

## 4. Discussion

Plant diseases caused by fungi destroy or contaminate a significant proportion of global agricultural production, making fungi the most deleterious class of plant pathogens [27]. Onion bulb rot is an economically significant disease that has emerged in Israel and other countries. This study identified different *Fusarium* species as the causal agent, and it adapted tools to identify the fungi and assay isolate virulence, as well as enable the screening of onion germplasm for resistance. The morphological characteristics of *Fusarium* spp. have been used in the past as the preferred methods for species identification [28]. However, the *Fusarium* genus is complex, and morphological differences may be challenging to observe. Therefore, DNA analysis is needed for accurate identification and characterization of the species. Isolates of the *Fusarium* spp. obtained from rotting bulbs of onion plants in commercial fields in northeastern Israel could not be differentiated morphologically from *Fusarium* strains found in other parts of Israel [1], and in other countries as well [25,26]. The DNA-sequence-based approach identified four *Fusarium* species among our field-collected samples: *F. proliferatum*, *F. oxysporum* f. sp. *cepae*, and two other species (less familiar as onion bulb rot causal agents), *F. acutatum* and *F. anthophilium*. The identity of the four species as the causal agents of onion bulb rot and onion seedlings disease was predominantly approved here using Koch’s postulates. This agrees with a previous report that the white strain of *F. proliferatum* is the most virulent causal agent in southern Israel, causing apperent symptoms in onion seedlings [1].

Since different *Fusarium* species were isolated from two different fields (in the north and south of the Golan Heights) and from two white and red onion cultivars (Table 2), each area or cultivar may harbor a unique mycobiome, as illustrated by the phylogenetic tree (Figure 6). Moreover, the onion bulb pathogenicity assay suggests that variation exists in disease sensitivity between cultivars. For example, Noam cv. being more resistant to some of the onion basal rot disease-causing agents, than Riverside cv. (Figure 9). It should be taken into consideration that several other factors may be involved in the two unique mycoflora separations. For example, the identification of different species in the two cultivars may be related to geographic factors (soil conditions, environmental conditions). Alternatively, anthocyanins and other pigments may influence plant–pathogen interactions, as well as disease resistance. These interesting variations should be examined more thoroughly in future studies. The seedling pathogenicity assay identified all *Fusarium* isolates as being able to significantly impact seed germination rate, hypocotyl elongation, and fresh biomass.

The increase in germination percentages in the presence of some of the isolates in the plate assay (B8, B14, and B16) may not reflect well the emergence inhibition in the plates. As seen clearly in Figure 7, there are variations among the seeds of the same treatment. Some have a very small radicle with no hypocotyl. In contrast, others have a well-developed radicle and hypocotyl. Thus, the hypocotyl elongation and fresh biomass better represent the influence of the pathogen under these conditions. Still, the interactions between those species should be studied to identify possible competition, antagonism, or synergism, as demonstrated in other cases [29]. Some of these *Fusarium* species may be less competitive pathogens or secondary parasites. Understanding the relationships within the fungal community involved in onion basal rot is essential for developing a preventive intervention program and should be examined more closely in subsequent work.

Onion bulb basal rot caused by the soil fungi *F. oxysporum* and *F. proliferatum* presents an increasingly severe threat to onion production in Israel [1]. A possible reason for the increased incidence of the disease in recent years could be the use of infected seeds for sowing. These two species were present in a substantial proportion of onion samples with basal rot originating from different continents [12]. Nevertheless, today there are only a few methods to control *Fusarium* basal rot diseases in commercial fields in Israel. Such practices include soil solarization [30], four-year crop rotation for onions, including avoidance of other members of the lily family, and breeding alliaceous crops for pest resistance [3]. In addition, using fludioxonil, thiophanate methyl [9], or metam sodium (formally a dithiocarbamate used for soil disinfection) as a pre-planting dip can provide some protection. However, contaminated agricultural equipment (e.g., plows) and workers facilitate disease spread to new areas. Indeed, the spread of the disease in Israel is unequivocally increasing, with reports of the fungus and disease symptoms in new agricultural regions with no previous disease record.

The onion seedling and bulb pathogenicity assay that was adjusted and used in the current work may be useful for breeding onion cultivars resistance and for screening fungicides, as was done in other crop fungal diseases (see, for example, [31]). Also, this assay may enable us to examine other treatments and to study the penetration and establishment stages of the pathogenesis. A scientific program dedicated to developing disease control cannot be established exclusively on field experiments during the growing season. This is due to the considerable investment involved in such experiments, the long period until results are received, and the fluctuations in environmental conditions that cause inconsistency in the results. The difficulty in receiving consistent and repeatable results is enhanced by the non-uniform, scattered nature of the *Fusarium* onion basal rot disease. Both the onion seedlings and bulbs pathogenicity assays offer a fast, reliable, and consistent solution to this problem. It is important to combine those two assays in order to achieve a more accurate prognostication about the aggressiveness of the pathogens involved in the onion basal rot disease. Despite these diagnostic methods’ advantages, they should be considered in a critical manner. Controlled-condition assays and even a seedlings pot assay are, in many cases, inconsistent in their ability to predict results in the field [32]. Still, they are necessary preliminary steps in ruling out ineffective treatments and in choosing the ones having the highest probability to succeed. A pot assay for screening fungicides to control the onion basal rot disease should be the focus of a subsequent study.

Infected onions without any previous symptoms in the field were also reported [13], suggesting post-harvest disease development. Indeed, *F. oxysporum* f. sp. *cepae* caused basal rot during lengthy storage [12]. Moreover, *Fusarium* infections result in contamination with mycotoxins, some of which have a notable impact on human and animal health [33]. The two most prevalent *Fusarium spp.* on diseased onion bulbs reported in this and previous studies [1], *F. oxysporum* and *F. proliferatum*, can produce mycotoxins such as enniatins, fumonisins, moniliformin, and fusaric acid [6]. Thus, the gradual increase in the incidence of onion yield loss due to *Fusarium* basal rot disease urgently requires the development of new approaches to restrict the pathogen outburst and spread.

The two above-mentioned *Fusarium* spp., *F. proliferatum* (isolated from white onion, Riverside cv.) and *F. oxysporum* f. sp. *cepae* (isolated from 565/505 cv., a newly developed red onion), are known as primary causal agents of basal rot, and given their pathogenic interaction with the onion found in this research, we can assume that these fungi are significant contributors to the disease. Regarding the other two *Fusarium* species identified in the diseased onion samples, *F. acutatum* and *F. anthophilium*, it is still not clear to what extent these species can exclusively cause disease in onions, or if the onion is merely used as an alternative host. Nevertheless, all four *Fusarium* species were pathogenic to the onion cultivar inspected here.

This study on the mycoflora that causes onion basal rot disease in northeastern Israel has implications beyond the local scale and may contribute to onion producers in other regions. Others can follow the methodology used here. It may help to distinguish between the *Fusarium* species involved, which can be challenging due to the rapid changes of this particular pathogen genus. It would be important to expand the research on the threat caused by *Fusarium* on onion to other plant species in different countries and continents.

## 5. Conclusions

In the current study, the fungal pathogens involved in onion basal rot disease in northeastern Israel were identified and characterized. Four *Fusarium* spp. species were identified in the contaminated field samples, *F. proliferatum*, *F. oxysporum* f. sp. *cepae*, and two species, less familiar as causative agents of this disease, *F. acutatum*, and *F. anthophilium*. Phylogenetic analysis revealed that these isolates were subdivided into two populations, a northern group that is isolated from white (Riverside cv.) onion bulbs and a southern group isolated from red (565/505 cv.) bulbs. Pathogenicity trials were conducted on onion seedlings and bulbs under controlled conditions to complete Koch’s postulates. These assays were able to identify virulence differences among the *Fusarium* species identified as being the causal agents of the disease in this country’s region. The pathogenicity assays also proved to be a diagnostic tool for determining the degree of sensitivity of different onion cultivars to the disease. The work findings are a necessary step towards uncovering the mycoflora of the diseased onion plants and developing a preventive program that will reduce the disease damage.

## Figures and Tables

**Figure 1 biology-09-00069-f001:**
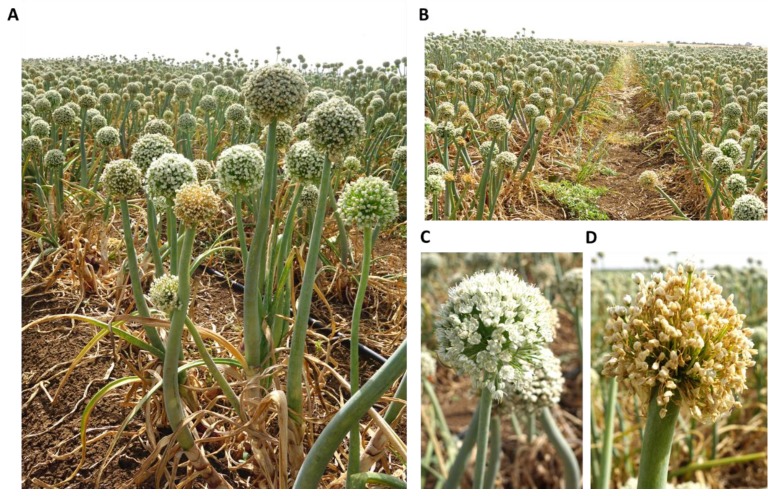
*Allium cepa* (white onion, Riverside cv.) flower symptoms. The commercial field is located near Kibbutz Ortal (northern Golan Heights, northeastern Israel) and photographed on 30 August 2017. The dehydrated inflorescences were scattered in the field (**A**,**B**). While the healthy plants had white inflorescences (**C**). The diseased, dried, plant had a yellowish to brown color inflorescences (**D**).

**Figure 2 biology-09-00069-f002:**
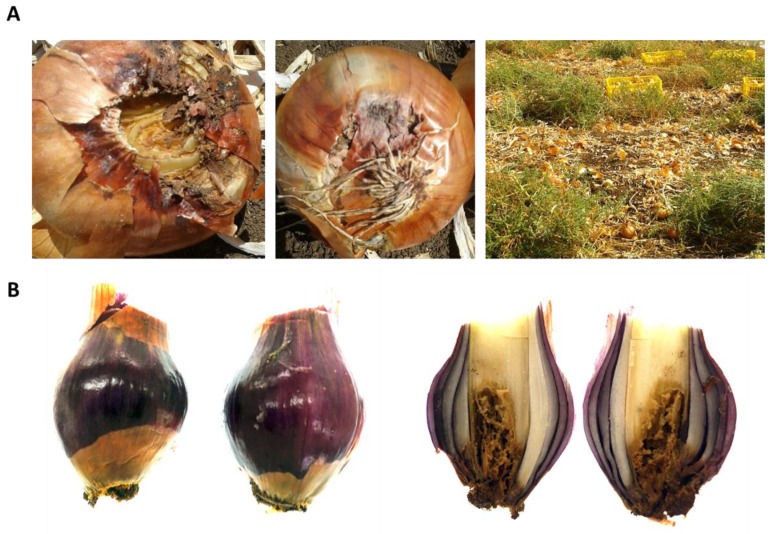
Onion bulb symptoms. (**A**) Basal rot symptoms appear on the Riverside cv. on the harvest day (30 August 2017, approximately 80 days from sowing) in the commercial field, presented in Figure 1. The two pictures on the left are a closeup of the symptoms, and the 3rd image is a wide view of the field from which these bulbs were collected. The density of the onion bulbs in the area may have facilitated disease spread. (**B**) Symptoms of basal rot on red onion, 565/505 cv., a newly developed cultivar from Hazera Seeds Ltd., Israel, collected from a commercial field near Moshav Eliad (southern Golan Heights, northeastern Israel) on the harvest day on 23 May 2018 (approximately 80 days from sowing).

**Figure 3 biology-09-00069-f003:**
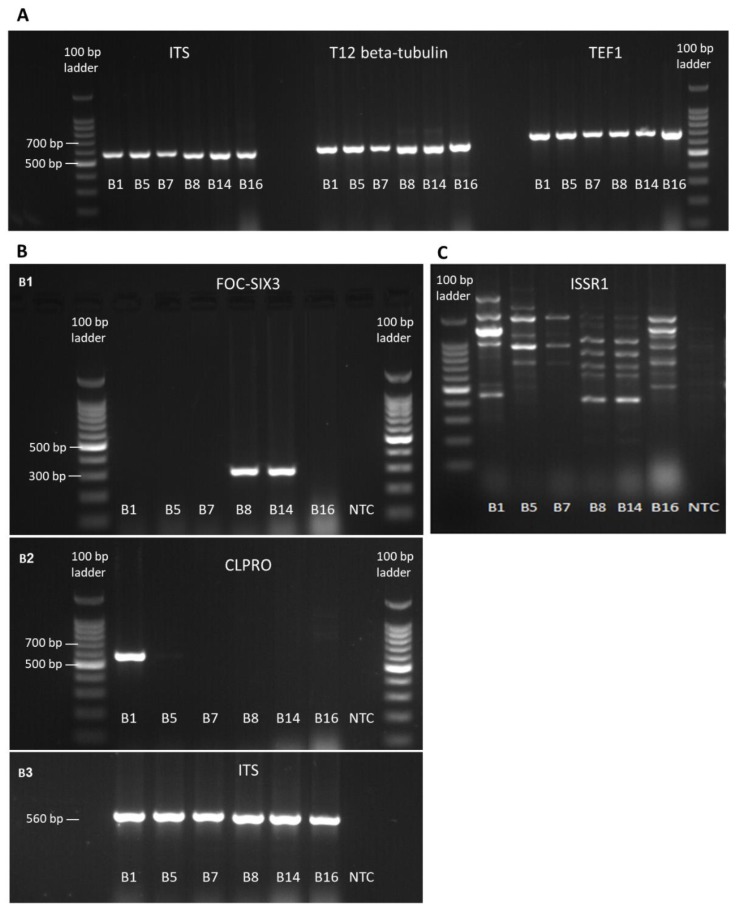
Molecular identification of the *Fusarium* spp. isolates. Molecular identification of *Fusarium* species by PCR amplification of the small subunit ribosomal RNA gene, internal transcribed spacer (ITS), the T12 beta-tubulin gene, the translation elongation factor-1 alpha (TEF1) gene (**A**), the *Fusarium oxysporum* f. sp. *cepae* (FOC) secreted in xylem genes 3 (SIX3) (**B1**), and the *Fusarium proliferatum* partial calmodulin gene (CLPRO) (**B2**). In the last two target genes, the ITS was used as a positive control (**B3**). Complex band profile (DNA fingerprinting) generated using the ISSR primers (**C**). Primer sets and reference to the sequences used are given in Table 1. NTC—no-template negative control in which sterile double-distilled water (DDW) was added instead of the DNA template.

**Figure 4 biology-09-00069-f004:**
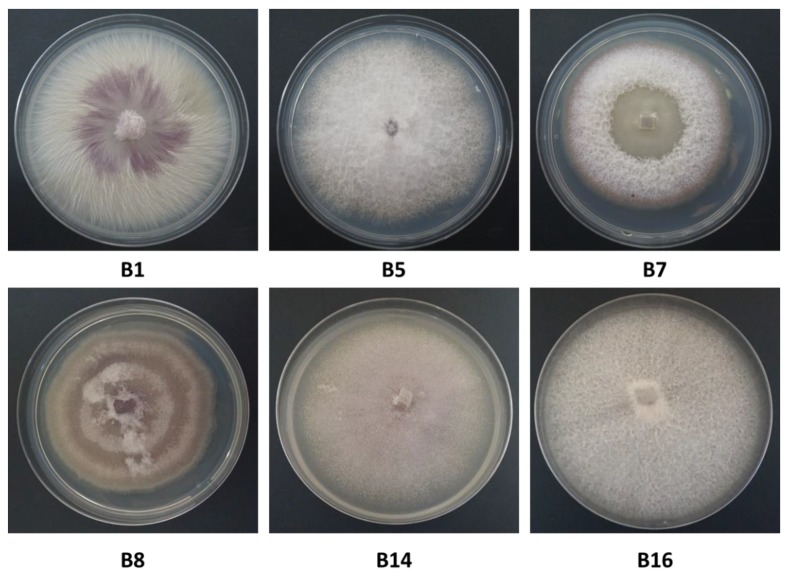
Colony morphology of the *Fusarium* spp. isolates. Strain designation and data are given in Table 2. Colonies were grown on potato dextrose agar (PDA; Difco Laboratories, Detroit, MI, USA) medium (prepared from the same batch) for seven days at 28 ± 1 °C in the dark.

**Figure 5 biology-09-00069-f005:**
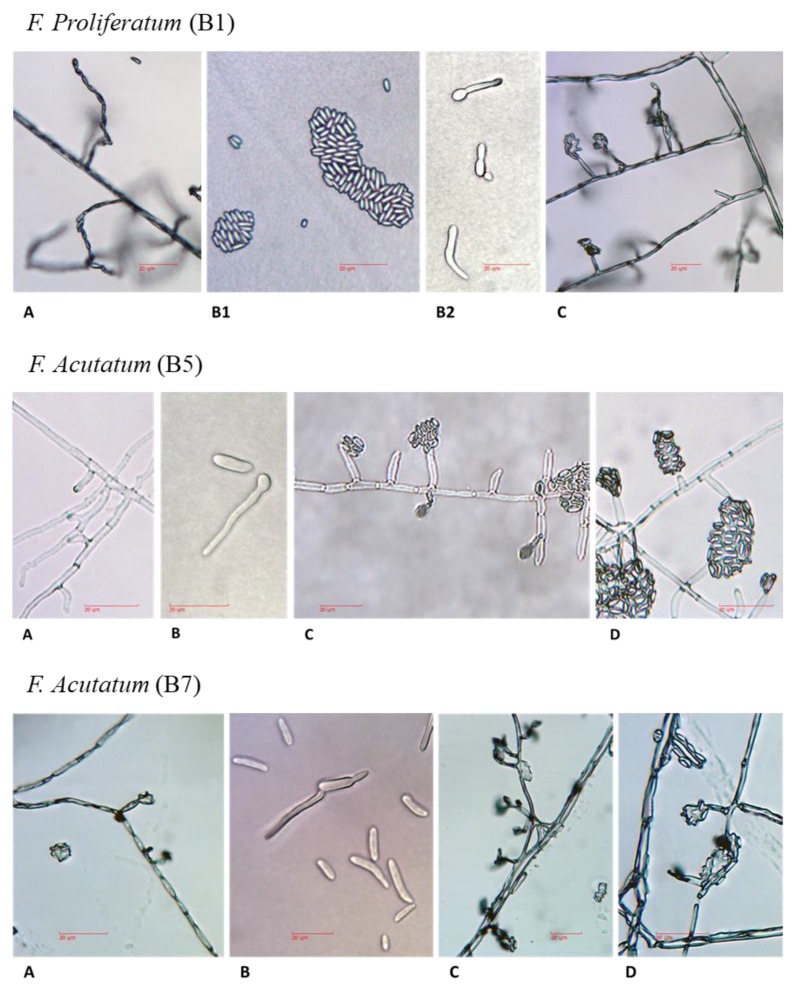
Microscopic characterization of the *Fusarium* spp. isolates. The *Fusarium* isolates grew on PDA medium for five days at 28 ± 1 °C in the dark. The slide culture technique was used [15]. Samples were studied using a light microscope (magnification 250:1, no stain used). (**A**) Hyphae. (**B**) Microconidia (small cells) and macroconidia (large cells). Conidia germinated via monopolar or bipolar germ tubes. (**C**) Conidiophores (monophialide) carry large quantities of microconidia on its false head. (**D**) Magnification of the conidiophores. Scale bars represent 20 µm.

**Figure 6 biology-09-00069-f006:**
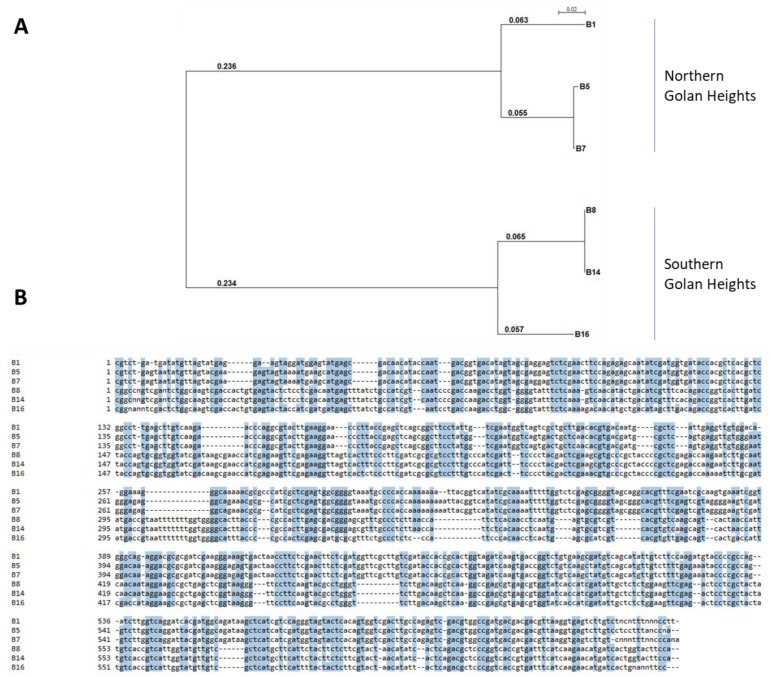
Phylogenetic analysis of the *Fusarium* isolates. Phylogenic relationships of the TEF1 gene and the six *Fusarium* isolates (presented in Table 2). The scale in the upper part refers to the genetic similarity of the isolates. Sequences were aligned using the Clone Manager 10.0 program (Sci Ed Software, Durham, NC, USA). The SeaView version 5.0 program (http://doua.prabi.fr/software/seaview) was used to generate a phylogenetic tree. The alignment was performed using bootstrap with 1000 replicates and excluded positions with gaps.

**Figure 7 biology-09-00069-f007:**
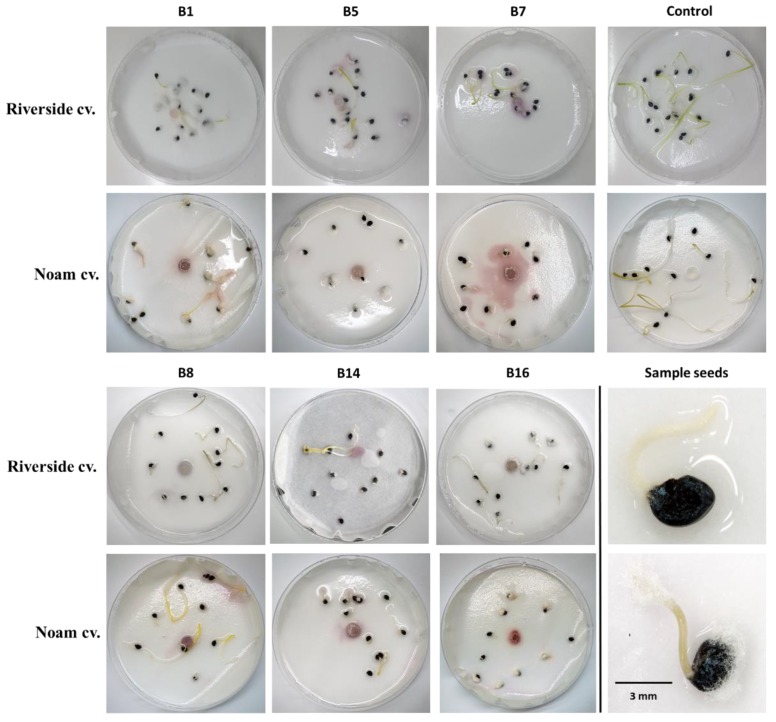
Seedling pathogenicity assay. Onion seeds of the Riverside (Orlando) and Noam cultivars were inoculated with six *Fusarium* isolates—B1, B5, B7, B8, B14, B16 (Table 2). Each group of 10-15 germinating seeds was inoculated with a 6-mm-diameter disc from a 5-day *Fusarium* sp. colony placed in the middle of the petri dish. A sterile 6-mm-diameter PDA disc was added to the control group. Sample seeds photos show minor (upper photo) or massive (lower photo) *Fusarium* white mycelial growth on or near the onion seeds. Photos were taken nine days after incubation at 28 ± 1 °C in the dark.

**Figure 8 biology-09-00069-f008:**
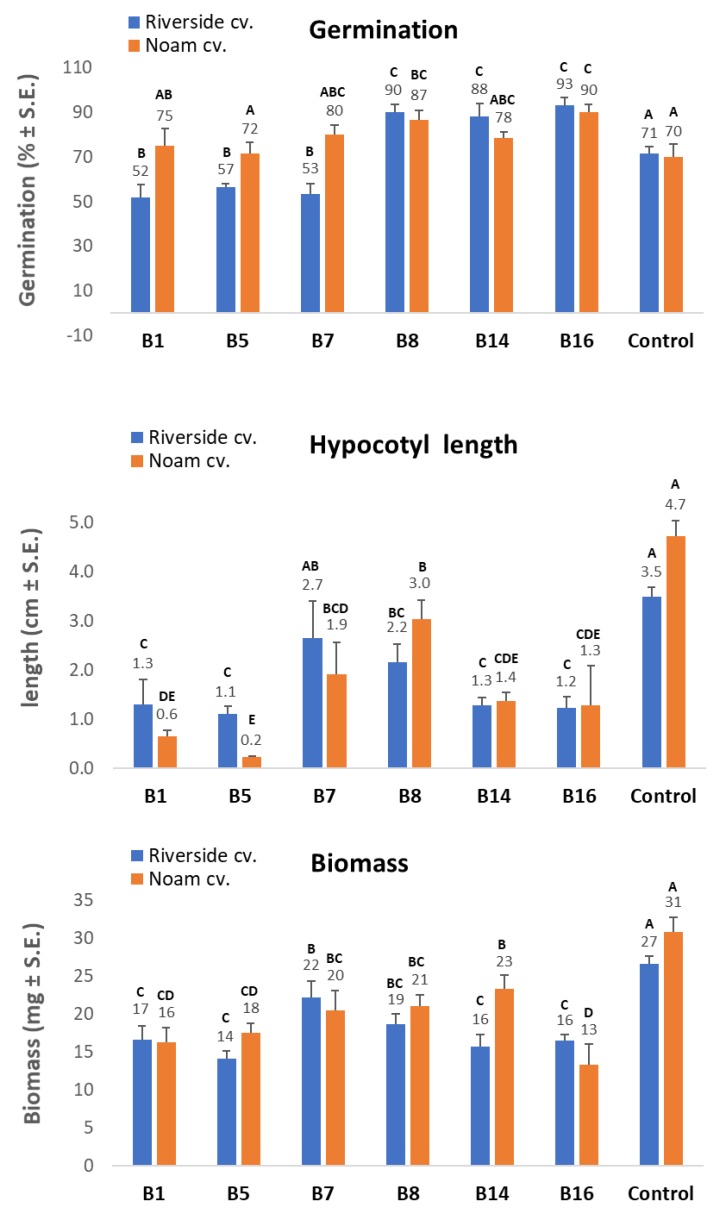
Growth of infected seedlings. The pathogenicity test using un-germinated seeds was conducted, as presented in Figure 7. The six *Fusarium* isolates were added to seeds *in vitro*, and the seedlings’ germination (damping-off) rate, hypocotyl elongation, and fresh biomass were measured nine days after incubation at 28 ± 1 °C in the dark. Vertical upper bars represent the standard error of the mean of six replications (Petri dishes, each containing 10-15 seeds). Different letters above the error bar indicate a significant difference (*p* < 0.05); the same letters indicate no significant difference.

**Figure 9 biology-09-00069-f009:**
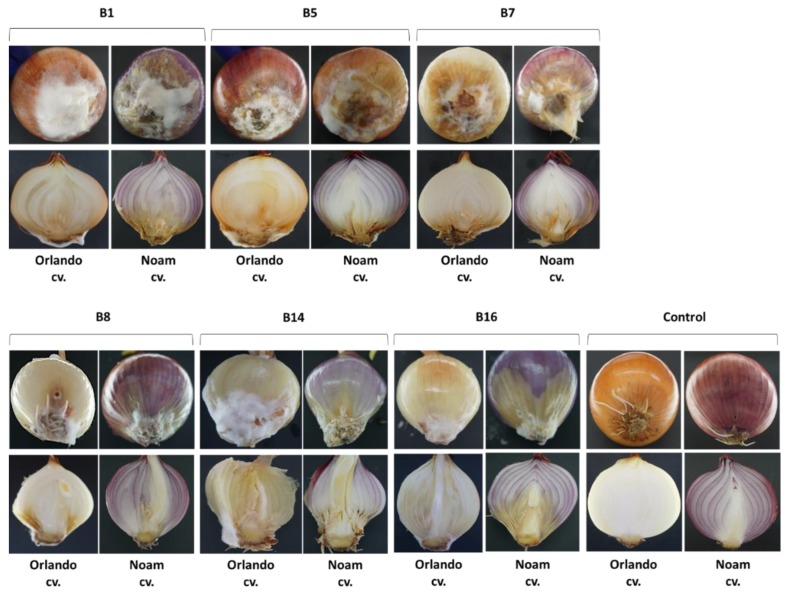
Onion bulb pathogenicity assay. Onion bulb inoculation assay was conducted on Riverside (Orlando) and Noam cultivars by injecting a conidial suspension of the six *Fusarium* isolates (Table 2) into the basal plate and incubating the bulbs in moisture bags in the dark at 28 ± 1 °C for two weeks. Upper panel—external symptoms on the onion basal plate. Lower panel—a cross-section of the bulbs. The disease symptoms included onion tissue decay, accompanied by the development of white hyphae on the external surface.

**Table 1 biology-09-00069-t001:** Primers used in this study for *Fusarium spp.* detection.

Primer	Sequence	Fragment Length	References
TEF1- *Fusarium*-specific (E1E2)	F-ATGGGTAAGGAGGACAAG	680 bp	[20]
	R-GGAAGTACCAGTGATCAT		
T12 beta-tubulin gene, fungi-specific	F-AACATGCGTGAGATTGTAAGT	580 bp	[21,22]
	R-TAGTGACCCTTGGCCCAGTTG		
ITS 1/4 universal DNA marker for fungi and plants	F-GGAAGTAAAAGTCGTAACAAGG	560 bp	[23]
	R-TCCTCCGCTTATTGATATGC		
*Fusarium oxysporum* f. sp. *cepae* (FOC) secreted in xylem genes 3 (SIX3)	F-ATGCGTTTCCTTCTGCTTATC	306 bp	This work
	R-AGGTGCGACATCAATGACAG		
*Fusarium proliferatum* partial calmodulin gene (CLPRO)	F-CTTTCCGCCAAGTTTCTTC	585 bp	[17]
	R-TGTCAGTAACTCGACGTTGTTG		
Inter simple sequence repeat (ISSR1)	F+R-AGAGAGAGAGAGAGA	Multiplay lengths	[18]

**Table 2 biology-09-00069-t002:** BLASTN identification of the *Fusarium* isolates from this study ^a^.

*Fusarium* spp.	Isolate	Gene ^b^	NCBI Accession	NCBI Score	Collection Site ^c^	Onion Cultivar ^d^
*F. proliferatum*	B1	TEF1	MK577936.1	99.85%	Kibbutz Ortal	Riverside (Orlando)
		BT2A	KC964149.1	99.44%		
		ITS	MN809262.1	100%		
*F. acutatum*	B5	TEF1	MK507814.1	100%	Kibbutz Ortal	Riverside ((Orlando)
		BT2A	U34431.1	100%		
		ITS	NR_111142.1	99.80%		
*F. acutatum*	B7	TEF1	MK507814.1	100%	Kibbutz Ortal	Riverside ((Orlando)
		BT2A	U34431.1	99.40%		
		ITS	NR_111142.1	99.81%		
*F. oxysporum* f. sp. *cepae*	B8	FOC-SIX3	KP964964.1	99.23%	Moshav Eliad	565/505
		TEF1	KP964881.1	99.55%		
		BT2A	MH827996.1	99.82		
		ITS	FJ605247.1	99.61%		
*F. oxysporum* f. sp. *cepae*	B14	FOC-SIX3	KP964964.1	100%	Moshav Eliad	565/505
		TEF1	KP964881.1	99.85%		
		BT2A	KJ433917.1	93.72%		
		ITS	FJ605247.1	98.82%		
*F. anthophilium*	B16	TEF1	LT996093.1	100%	Moshav Eliad	565/505
		BT2A	LT996114.1	99.45%		
		ITS	NR_111142.1	99.43%		

^a^ Strains B1, B5, and B7 were isolated on 30 August 2017; strains B8, B14, and B16 were isolated on 23 May 2018. ^b^ TEF1, translation elongation factor-1 alpha gene; BT2A, beta-tubulin gene; ITS, small subunit ribosomal RNA gene, internal transcribed spacer; FOC-SIX3, *Fusarium oxysporum* f. sp. *cepae* (FOC) secreted in xylem genes 3 (SIX3). ^c^ Kibbutz Ortal is located in the northern Golan Heights; Moshav Eliad is located in the southern Golan Heights. ^d^ Riverside (Orlando) is a white onion cultivar, (565/505) is a newly developed red onion cultivar. Both were supplied by Hazera Seeds Ltd., Berurim MP Shikmim, Israel.

**Table 3 biology-09-00069-t003:** Microscopic characterization and identification of *Fusarium* spp. isolates.

Isolate No.	*Fusarium* spp.	Colony Growth Rate (cm/day ± S.E.) ^a^	Microconidia (µm)	Macroconidia (µm)	Hyphal Width (µm)
**B1**	*F. proliferatum*	1.40 (±0.04)	6.2 × 2.9	20.5 × 3.2	3.6
**B5**	*F. acutatum*	1.24 (±0.04)	5.8 × 2.7	18.2 × 3.7	3.4
**B7**	*F. acutatum*	1.07 (±0.04)	9.75 × 2.4	52.7 × 3.1	3.3
**B8**	*F. oxysporum* f. sp. *cepae*	0.87 (±0.02)	7.4 × 3.3	20.5 × 3.9	3.2
**B14**	*F. oxysporum* f. sp. *cepae*	0.83 (±0.02)	7.5 × 3.7	20.9 × 3.9	3.1
**B16**	*F. anthophilium*	0.86 (±0.01)	7.2 × 3.1 (oblong)	18.8 × 4.5	3.3
			3.8 × 3.1 (elliptical)		

^a^ Isolates were grown on PDA for five days at 28 ± 1 °C in the dark. Values represent the average size calculated from six representative spores or hyphae.

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
