# Peer review of "Isolation and Identification of Fusarium spp., the Causal Agents of Onion (Allium cepa) Basal Rot in Northeastern Israel"

_biology, 2020, doi:10.3390/biology9040069_

Round 1
Reviewer 1 Report
The manuscript describes isolation and identification of onion basal rot causative agents in northern Israel. Four Fusarium spp. isolates were obtained of which F. proliferatum and F. oxysporum f. sp. cepae are known onion pathogens and F. acutatum and F. anthophilium are less frequently connected with basal rot. The fungus morphology and phylogeny were characterized. In addition, the pathogenicity and effects of isolated fungi for onion seeds and bulbs were confirmed in laboratory tests. The study increases the knowledge of pathogenicity of Fusarium spp. fungi and may help to develop protection against Fusarium-caused onion basal rot.
The structure of the manuscript needs some rearrangements, some language checking is needed and some details in figure legends and text need to be checked. In many paragraph there is unnecessary repetition. The results should be mentioned in results section not in material and methods and result interpretation should be in discussion section not in results or in figure legends. The onion bulb assay should produce some measurable data in order to be more repeatable method to measure sensitivity level. With proper corrections the manuscript could be recommended for publication.
Details of corrections:
Paragraph order: Change the names and move paragraphs (after suggested corrections below) in new order as follows and update the table and figure names:
Line 218: 3.1. Isolation and identification of pathogens from diseased onion plants
Line 218 (replacement): 3.1. Isolation of pathogens from diseased onion plants
Line 239: 3.2. Morphological and microscopic characterization and identification
Line 239 (replacement): 3.3. Morphological and microscopic characterization and identification
Line 240: 3.2.1. Morphology of plated cultures.
Line 240 (replacement): 3.3.1. Morphology of plated cultures
Line 257: 3.2.1. Microscopy
Line 257 (replacement): 3.3.1. Microscopy
Line 282: 3.3. Molecular DNA-based identification
Line 282 (replacement): 3.2. Molecular DNA-based identification
Line 173: move Table 2 to result into the paragraph “Molecular DNA-based identification”
Other suggestions for the text:
Line 29: Inoculating seeds with spores suspension
Line 29 (replacement): Inoculating seeds with spore suspension
Line 38: remove Allium cepa, basal rot, Fusarium, onion (these words are mentioned in the title)
Line 45: onion bulbs production
Line 45 (replacement): onion bulb production
Line 80: the problem considerably worsened
Line 80 (replacement): the problem is considerably worsened
Line 101: add the cultivars
Line154: add the used DNA amount in grams
Line 160-161: The PCR-amplified oligonucleotide products were run on 1.5% agarose gel concentration and electrophoresis
Line 160-161 (replacement): The PCR products were separated on 1.5% agarose gel electrophoresis
Line 163 Table 1: instead of the table the primers with their references could be listed in the text
Line 161-162: remove the sentence “The sequences were identified by BLASTN (NCBI), as detailed below and shown in Table 2.”
Line 181: change the title to “seedling pathogenicity assay”
Line 182 to 184: The pathogenicity test on onion seedlings was designed to measure the level of virulence of the Fusarium spp. isolates and was carried out in six replicates. The experiment was repeated twice and similar results were obtained in both cases.
Line 182 to 184 (replacement): The pathogenicity test on onion seedlings was designed to measure the level of virulence of the Fusarium spp. isolates. The test was repeated twice with six replicates.
Line 184 to 187: remove the sentence “The Riverside (Orlando) cv. (yellow onion) and Noam cv. (purple onion) seeds (supplied by Hazera Seeds Ltd., Berurim MP Shikmim, Israel) were inoculated with the six Fusarium isolates – B1, B5, B7, B8, B14, B16 (F. proliferatum, two F. acutatum isolates, two F. oxysporum f. sp. cepae and F. anthophilium, respectively, see Table 2).”
Line 187: The seeds were rinsed in DDW,
Line 187 (replacement): The Riverside (Orlando) cv. (yellow onion) and Noam cv. 184 (purple onion) seeds (supplied by Hazera Seeds Ltd., Berurim MP Shikmim, Israel) were rinsed in DDW,
Line 197 to 200: For the Koch’s postulates accomplishment, an onion bulbs pathogenicity assay was conducted on Riverside (Orlando) cv. and Noam cv. (white and purple onion cultivars, respectively) by injecting a conidial suspension of the Fusarium isolates (Table 2) into the basal plate and inoculating the bulbs in moisture bags.
Line 197 to 200 (replacement): For the Koch’s postulates accomplishment, an onion bulbs pathogenicity assay was conducted on Riverside (Orlando) cv. and Noam cv. (white and purple onion cultivars, respectively).
Lines 219 to 227: Six different isolates belonging to four different Fusarium species were isolated from diseased white and red onion bulbs collected randomly from commercial fields in the Golan Heights in northeastern Israel. The disease symptoms observed on onions plants sampled in this study were dehydration of the flowers (Figure 1) and bulb rot, spreading from the onion basal plate upwards in the scales, resulting in discolored and watery bulb tissue (Figure 2). We isolated single spore cultures from these field-collected fungi, and in the following sections, we report their identification and characterization. Following isolation, we artificially inoculated mature onion bulbs (see below) and re-isolated and confirmed the identity of the same fungi, thus accomplishing Koch's postulates for the identification of causal disease agents.
Lines 219 to 227 (replacement): The disease symptoms observed on onions plants sampled in this study were dehydration of the flowers (Figure 1) and bulb rot, spreading from the onion basal plate upwards in the scales, resulting in discolored and watery bulb tissue (Figure 2). We isolated single spore cultures from these field-collected fungi and six different isolates were isolated from diseased white and red onion bulbs collected randomly from commercial fields in the Golan Heights in northeastern Israel.
Line 249: cultured
Line 249 (replacement): cultures
Line 274: Figure 4 legend: What are D pictures? Is the magnification same in every picture? The scale bars are different sizes. Is this because of modified picture size?
Line 283: Followed the microscopic characterization, final confirmation of the pathogen
Line 283 (replacement): Confirmation of the pathogen
Line 300: Figure 5 legend: use TEF in the picture instead of E1E2
Line 328: Onion seedling assay developed as
Line 328 (replacement): Onion seedling assay was developed as
Line 329 to 333: remove: This is essential since a seedling pathogenicity assay may enable us to examine different fungicides and other treatments to locate resistant onion cultivars and to study the penetration and establishment at the pathogenesis stages. The assay comprises sowing the seeds on under moisture conditions on Watman paper in a Petri dish and adding a Fusarium colony agar disk to the center of the plate.
Line 335 to 340: Adding the F. proliferatum (isolate B1) or F. acutatum (B7, B5) isolates to onion seeds (cultivars Riverside) under controlled conditions in Petri dishes significantly suppressed seed germination after nine days compared to the control (P < 0.05, Figure 8).No difference in the damping-off effect was found between F. oxysporum f. sp. cepae (B8/B14) or F. anthophilium (B16) and the control. The Noam cv. seeds' germination was not affected by the inoculation with the Fusarium isolates.
Line 335 to 340 (replacement, based on the figure’s significant difference labels?): Presence of F. proliferatum (isolate B1) or F. acutatum (B7, B5) significantly suppressed and presence of F. oxysporum f. sp. cepae (B8/B14) or F. anthophilium (B16) significantly increased seed germination of onion seeds (cultivars Riverside) under controlled conditions in Petri dishes after nine days compared to the control (P < 0.05, Figure 8). No difference in the damping-off effect was found between F. oxysporum f. sp. cepae (B8/B14) or F. anthophilium (B16) and the control. The Noam cv. seeds' germination was not affected by the inoculation with the Fusarium isolates B1, B5, B7 and B14. However, the germination was significantly enhanced by the isolates B8 and B16.
Line 360 to 362: remove: Since the wounding of the plant tissue facilitates infection, we injected a conidial suspension of the Fusarium isolates into the basal plate and incubated the bulbs in moisture bags.
Line 367 to 371: What is the statement from bulb experiment based on? This part needs more explanation or need to be removed!: The onion bulb assay was capable of revealing the degree of sensitivity of the two onion cultivars to the disease, Noam being less sensitive to the pathogen than Riverside. In the seedling pathogenicity assay, the seed germination results of F. proliferatum (isolate B1) and F. acutatum (B7, B5) isolates supported this conclusion. Nevertheless, the fresh biomass and epicotyl elongation of the inoculated seeds were more ambiguous in determining the degree of sensitivity of the onion cultivars.
Line 380: remove: Noam bulbs were less sensitive to the pathogen than Riverside bulbs.
Line 457 to 471: need to be compressed. Do not repeat the abstract say only the main conclusions.
Author Response
Responses to Reviewer 1’s comments
We are thankful for the reviewer for investing substantial work that contributes significantly to this manuscript. The many helpful and essential remarks and suggestions improved this scientific paper remarkably and made it more accurate, clear, focused and well-structured. Your contribution is greatly appreciated.
General comments:
The structure of the manuscript needs some rearrangements, some language checking is needed and some details in the figure legends and text need to be checked. In many paragraphs there is unnecessary repetition. The results should be mentioned in the results section and not in materials and methods, and results interpretation should be in the discussion section not in the results or in figure legends.
We made our best effort to address this issue. We believe that after integrating the remarks and suggestions of all four reviewers, the manuscript’s scientific accurateness, structure and focus were significantly improved.
A professional English scientific copy editor edited the entire revised manuscript.
The onion bulb assay should produce some measurable data in order to be a more repeatable method to measure the sensitivity level.
Indeed, but providing accurate, measurable data from this specific pathogenicity assay is truly challenging. Therefore, we used the seedling assay to give the quantifiable parameters and the onion bulb assay to mimic the situation in the field and provide more qualitative results. A similar seedling test was used by other researchers for the same reason (see, for example, Palmero et al. (2012) Fusarium proliferatum isolated from garlic in Spain: identification, toxigenic potential and pathogenicity on related Allium species. Phytopathologia Mediterranea, 207-218).
A repeatable method to measure the sensitivity level is indeed an important aspect that should be developed more and be the focus of a different study as was done by other researchers (for example, Wang et al. (2019). Pathogenic Fusarium oxysporum f. sp. cepae growing inside onion bulbs emits volatile organic compounds that correlate with the extent of infection. Postharvest Biology and Technology 152:19-28).
Details of corrections:
Paragraph order: Change the names and move paragraphs (after suggested corrections below) in a new order as follows and update the table and figure names.
We accepted all of the changes suggested by the reviewer and made the corrections as detailed below.
Line 218: 3.1. Isolation and identification of pathogens from diseased onion plants
Line 218 (replacement): 3.1. Isolation of pathogens from diseased onion plants
Corrected as advised.
Line 239: 3.2. Morphological and microscopic characterization and identification
Line 269 (replacement): 3.3. Morphological and microscopic characterization and identification
Corrected as advised.
Line 240: 3.2.1. Morphology of plated cultures
Line 270 (replacement): 3.3.1. Morphology of plated cultures
Corrected as advised.
Line 257: 3.2.1. Microscopy
Line 287 (replacement): 3.3.2. Microscopy
Corrected as advised.
Line 282: 3.3. Molecular DNA-based identification
Line 134 (replacement): 2.2. Molecular DNA-based identification
Corrected as advised.
Line 174: move Table 2 into the paragraph “Molecular DNA-based identification”
Corrected as advised.
Other suggestions for the text:
Line 29: Inoculating seeds with spores suspension
Lines 29-30 (replacement): Inoculating seeds with a spore suspension
Corrected as advised.
Line 38: remove Allium cepa, basal rot, Fusarium, onion (these words are mentioned in the title)
The terms were deleted from the Keywords and replaced with other words that are not mentioned in the title.
Line 45: onion bulbs production
Line 46 (replacement): onion bulb production
Corrected as advised.
Line 80: the problem considerably worsened
Line 82 (replacement): the problem is considerably worsened
Corrected as advised.
Line 101: add the cultivars
The following sentence was added as advised (lines 103-104): “The study focused on two varied common crops, Riverside (Orlando) cv. (white onion) and Noam cv. (purple onion) (supplied by Hazera Seeds Ltd., Berurim MP Shikmim, Israel).”
Line154: add the used DNA amount in grams
The amount of Fusarium isolates’ DNA used in the PCR reaction mixture as a template was not measured. However, the DNA extraction using the Master Pure Yeast DNA Purification Set Kit (Sigma) yields a high and quality amount of DNA, as clearly expressed in the PCR results (Figure 3). We have used this procedure routinely in our lab for many years, with satisfying and repeatable results.
Lines 160-161: The PCR-amplified oligonucleotide products were run on 1.5% agarose gel concentration and electrophoresis
Lines 171-172 (replacement): the PCR products were separated on 1.5% agarose gel electrophoresis
Corrected as advised.
Line 163, Table 1: instead of the table, the primers with their references could be listed in the text
This is true, but we think that presenting the data as a table facilitates finding them and extracting the sequences for subsequent future studies.
Lines 161 to 162: remove the sentence “The sequences were identified by BLASTN (NCBI), as detailed below and shown in Table 2.”
The sentence was removed as advised.
Line 186: change the title to “Seedling pathogenicity assay”
Corrected as advised.
Lines 182 to 184: The pathogenicity test on onion seedlings was designed to measure the level of virulence of the Fusarium spp. isolates and was carried out in six replicates. The experiment was repeated twice and similar results were obtained in both cases.
Lines 187 to 188 (replacement): The pathogenicity test on onion seedlings was designed to measure the level of virulence of the Fusarium spp. isolates. The test was repeated twice with six replicates.
Corrected as advised.
Lines 184 to 187: remove the sentence “The Riverside (Orlando) cv. (yellow onion) and Noam cv. (purple onion) seeds (supplied by Hazera Seeds Ltd., Berurim MP Shikmim, Israel) were inoculated with the six Fusarium isolates – B1, B5, B7, B8, B14, B16 (F. proliferatum, two F. acutatum isolates, two F. oxysporum f. sp. cepae and F. anthophilium, respectively, see Table 2).”
The sentence was removed as advised.
Line 187: The seeds were rinsed in DDW,
Line 189 (replacement): The Riverside (Orlando) cv. (yellow onion) and Noam cv. seeds were rinsed in DDW,
Corrected as advised.
Line 104: Riverside (Orlando) cv. (white onion) and Noam cv. 200 (purple onion) (supplied by Hazera Seeds Ltd., Berurim MP Shikmim, Israel).
The information of the seed supplier was added in line 104, where the two cultivars were first mentioned in the text.
Lines 197 to 200: For the Koch’s postulates accomplishment, an onion bulb pathogenicity assay was conducted on Riverside (Orlando) cv. and Noam cv. (white and purple onion cultivars, respectively) by injecting a conidial suspension of the Fusarium isolates (Table 2) into the basal plate and inoculating the bulbs in moisture bags.
Lines 200 to 201 (replacement): For the Koch’s postulates accomplishment, an onion bulb pathogenicity assay was conducted on Riverside (Orlando) cv. and Noam cv. (white and purple onion cultivars, respectively).
Corrected as advised.
Lines 219 to 227: Six different isolates belonging to four different Fusarium species were isolated from diseased white and red onion bulbs collected randomly from commercial fields in the Golan Heights in northeastern Israel. The disease symptoms observed on onions plants sampled in this study were dehydration of the flowers (Figure 1) and bulb rot, spreading from the onion basal plate upwards in the scales, resulting in discolored and watery bulb tissue (Figure 2). We isolated single spore cultures from these field-collected fungi, and in the following sections, we report their identification and characterization. Following isolation, we artificially inoculated mature onion bulbs (see below) and re-isolated and confirmed the identity of the same fungi, thus accomplishing Koch’s postulates for the identification of causal disease agents.
Lines 221-225 (replacement): The disease symptoms observed on onion plants sampled in this study were dehydration of the flowers (Figure 1) and bulb rot, spreading from the onion basal plate upwards in the scales, resulting in discolored and watery bulb tissue (Figure 2). We isolated single-spore cultures from these field-collected fungi, and six different isolates were isolated from diseased white and red onion bulbs that were collected randomly from commercial fields in the Golan Heights in northeastern Israel.
Corrected as advised.
Line 249: cultured
Line 269 (replacement): cultures
Corrected as advised.
Line 274: Figure 4 legend: What are D pictures? Is the magnification the same in every picture? The scale bars are different sizes. Is this because of modified picture size?
D in Figure 4 is a magnification of the conidiophores. We added this information to the figure legend.
Indeed, the scale bars are of different sizes because of the modified picture size. All the scale bars represent 20 µm.
Line 283: Followed the microscopic characterization, final confirmation of the pathogen
Line 240 (replacement): Confirmation of the pathogen
Corrected as advised.
Line 304: Figure 5 legend: use TEF in the picture instead of E1E2
Corrected as advised.
Table 1, Figure 5, and the relevant associated text were updated and now include new PCR data (the Fusarium proliferatum partial calmodulin gene identification) that provide additional strength and support to the findings presented in the previous manuscript version about the identity of the Fusarium isolates.
Line 328: Onion seedling assay developed as
Line 328 (replacement): Onion seedling assay was developed as
The sentence was modified according to the request of Reviewer 2.
Lines 329 to 333: remove: This is essential since a seedling pathogenicity assay may enable us to examine different fungicides and other treatments to locate resistant onion cultivars and to study the penetration and establishment at the pathogenesis stages. The assay comprises sowing the seeds on under moisture conditions on Watman paper in a petri dish and adding a Fusarium colony agar disk to the center of the plate.
The sentence was removed as advised. Part of this sentence was integrated into the Discussion (lines 442-443): “Also, this assay may enable us to examine other treatments and to study the penetration and establishment stages of the pathogenesis.”
Lines 335 to 340: Adding the F. proliferatum (isolate B1) or F. acutatum (B7, B5) isolates to onion seeds (Riverside cultivars) under controlled conditions in petri dishes significantly suppressed seed germination after nine days compared to the control (P < 0.05, Figure 8). No difference in the damping-off effect was found between F. oxysporum f. sp. cepae (B8/B14) or F. anthophilium (B16) and the control. The Noam cv. seeds’ germination was not affected by the inoculation with the Fusarium isolates.
Lines 340 to 346 (replacement, based on the Figure’s significant difference labels): The presence of F. proliferatum (isolate B1) or F. acutatum (B7, B5) significantly suppressed, and the presence of F. oxysporum f. sp. cepae (B8/B14) or F. anthophilium (B16) significantly increased, the seed germination of onion seeds (cultivars Riverside) under controlled conditions in petri dishes after nine days compared to the control (P < 0.05, Figure 8). No difference in the damping-off effect was found between F. oxysporum f. sp. cepae (B8/B14) or F. anthophilium (B16) and the control. The Noam cv. seeds’ germination was not affected by the inoculation with the Fusarium isolates B1, B5, B7 and B14. However, the germination was significantly enhanced by the isolates B8 and B16.
The paragraph was corrected as advised. Indeed, the suggested sentence described more accurately the results and the significant difference labels.
Lines 360 to 362: remove: Since the wounding of the plant tissue facilitates infection, we injected a conidial suspension of the Fusarium isolates into the basal plate and incubated the bulbs in moisture bags.
The sentence was removed as advised.
Lines 367 to 371: What is the statement from bulb experiment based on? This part needs more explanation or needs to be removed! The onion bulb assay was capable of revealing the degree of sensitivity of the two onion cultivars to the disease, Noam being less sensitive to the pathogen than Riverside. In the seedling pathogenicity assay, the seed germination results of F. proliferatum (isolate B1) and F. acutatum (B7, B5) isolates supported this conclusion. Nevertheless, the fresh biomass and epicotyl elongation of the inoculated seeds were more ambiguous in determining the degree of sensitivity of the onion cultivars.
The paragraph was rewritten and shortened to add more clarity and accuracy (lines 374-376): “Based only on the qualitative estimation presented here, the onion bulb assay may reveal the degree of sensitivity of the two onion cultivars to the disease. According to this test in the F. oxysporum f. sp. cepae (B8/B14) injection, the Noam cv. symptoms are less severe than the Riverside cv. symptoms.”
Line 380: remove: Noam bulbs were less sensitive to the pathogen than Riverside bulbs.
The sentence was rephrased to: “According to this test in the F. oxysporum f. sp. cepae (B8/B14) injection, the Noam cv. symptoms are less severe than the Riverside cv. symptoms.” (lines 375-376).
Lines 457 to 471: need to be compressed. Do not repeat the abstract, say only the main conclusions.
The Conclusion section was shortened, as suggested, and now focuses on summarizing the Results and Discussion.
Reviewer 2 Report
This is a very interesting manuscript with sound experimental design. Although reasonably well written, some concerns need to be addressed.
First, there are a number of grammatical issues (as well as some formatting issues, such as periods in the wrong place) and awkward sentences. There are too many to list, but these need to be addressed in order to provide clarity. The authors would do well to seek feedback from a native English speaker to help with these. In some cases the ideas need to be better articulated, and this could be at least partially addressed with improvements to language usage.
Second, the objectives need to be more clearly stated; at some point it seems to the reader that the authors may have developed methods, though it is not clear whether this is in fact the case. If so, this needs to be integrated into the introduction as an objective, clearly stated in the methods, and needs to better articulated and discussed in the discussion/conclusions. What are the pre-existing methods, and what advantages can be brought by this method. Additionally, some comments are offered about fungicide screening, but this was not done in the current work, and the discussion on this topic is not very convincing simply because it was not properly articulated.
Finally, the authors suggest that species difference may be related to cultivar difference; however, their are several other possibilities that have not been discussed. For example, the identification of different species on the two cultivars may be related to geographic considerations (soil conditions, environmental conditions), or a preference for different colors--anthocyanins and other pigments may influence plant-pathogen interactions as well as disease resistances. I suspect more work is needed to answer these questions, but the discussion seems incomplete in this context.
Otherwise, very interesting manuscript and a pleasure to review. Please find some specific comments/concerns below.
Line 23 - Fusarium-specific genes or primers?
Line 23 - field samples of what? soils? onions? onion crop debri?
Line 48, 461 Fusarium species are widely distributed in what soil and associated with what plants?
Line 56 - F.oxysporum most damaging pathogen or plant pathogen?
Line 65 - personal communications with whom?
Line 85 - is it onion strains? or should it be cultivars/varieties?
Line 113 - recommend to say 'pure colony' rather than clean.
Line 122 - please clarify what is meat by single source
Line 123 - obtained instead of received.
Line 127 - please specify media
Line 134 - cover slip or glass slide?
Microscopy - it is not clear whether the 10 uL drops of conidia in PDB or DDW was used to inoculate nutrient agar. This paragraph should be revisited for clarification on the steps used.
Line 142 - which TEF gene (ie. 1- alpha, or another kind?); are the primers specific to Fusarium oxysporum species? or specific to different Fusarium species?
Line 160 - was electrophoresis for size determination or for gel purification prior sequencing? or both?
Table 1 - F- is on separate line from primer for one primer; typo - multiplay lengths?
Riverside (Orlando) cultivar has been described as white or yellow in different places in the text
Section 2.5 - please clarify what are the 6 replications? petri dishes with 15 seeds each?
Section 2.5 - Is this a seed assay or seedling assay?
Line 189 - Whatman spelling, here and elsewhere in the manuscript
SEction 2.6 - Onion bulb inoculation assay? or pathogenicity assay?
Line 199 - bulbs were inoculated in a moisture bag-- please specify how they were inoculated, with how much of what? Also, what is a moisture bag? in a moist bag?
Line 201 - biological hood? or biohazard hood? Was the entire assay carried out in the hood, or was set up in the hood?
Line 208-209 - how were disease symptoms evaluated?
Line 245 - isolates B1 ? or isolate B1?
Line 245 - F. proliferatum is a fast growing species--is this commonly known, or was this an observation made here. If the former, then a reference is needed, if the latter, perhaps the authors mean fastest growing among the species evaluated. Further to this, if this observation was made in a single isolate, than this could simply be a fast-growing isolate and not necessarily an F. proliferatum trait. Some clarity is needed in the writing.
Line 248 - colony color is not ideally evaluated in PDA as there can be variation among batches of media.
Line 249 - cultured? or cultures?
Table 3 - Fusarium spp. would be better description of the identification column. Please check spelling in table, and correct typos: Colony Growth; Hyphal Width. Please check that growth rates described in the text match that presented in the tables. There seems to be a disconnect in some of the values.
Figure 4 - description missing for 'D'
Section 3.2 title: identification of what?
section 3.3, title is awkward
Line 288 Gene Bank or GenBank?
Line 289, not clear what amplified sequence
Line 294 - Sentence is not clear and should be rephrased.
Once the acronyms have been defined at first use, do not need to spell them out again (ie. ITS, TEF), his goes for figures too.
Figure 5 - there is a redundant sentence regarding primers
Figure 5 this would be better as a supplementary file.
First sentence section 3.4 is awkward and should be rephrased.
Line 309, figure 6 A or B?
Line 311 - we're or were?
Line 316, 401 two white and red onion cultivars - two of each? one of each? not clear?
Line 316-17 - are the authors suggesting that the cultivar may influence the microflora? Or that the type of onion would do this? Please clarify.
Line 328 - onion seedling assay developed--where was this developed? If the method developed here that was not made clear.
Line 329-331 - This phrase is awkward. Suggest reworking this to better express ideas. Furthermore, this seems more of a discussion point than a result.
Figure 8 - Any comment on the higher germination rate of treatments in some cases
Line 341, 370 - epicotyl or hypocotyl
Figure 7 - if the seeds are not evenly spaced, could they have an interaction with each other an increase or decrease apparent pathogenicity? Mycelial growth on one can spread to another.
Line 386 - if this study developed tools, it is not clear in the manuscript what those tools are
Please check that the term isolate and species are used in the right places. For example, Line 396--should this be the identity of four species?
Noam is more resistant than Riverside. Is there any knowledge as to whether there are inherent difference in white vs red, or is this something that has not yet been explored?
Line 407 - states that all Fusarium isolates evaluated "significantly impact seed germination rate", In figure 8 some isolates increased germination rate. How can this be explained?
Line 409 - "in over cases" is there a number missing here?
Line 421 - is their an extra or missing word? "for" what?
Line 427 - useful for breeding -- as a selection tool? a screening tool? please clarify
Line 435 - prognostication about what?
Line 437-439 - discussion on treatments and how pot assays are an important first step, but the authors failed to first describe how these treatments could be applied in a pot assay, or if others have done this for onion
Line 455 " as will be detailed below" is a strange way to end a discussion. The discussion should be comprehensive in it's own right. Whereas the conclusion is more to summarize results and discussion.
Line 459 - "Rot in these plants", better to be specific here.
Author Response
Responses to Reviewer 2’s comments
We would like to express our deep appreciation to the reviewer for important and helpful corrections, suggestions and advice. The time and effort invested are greatly appreciated, and without a doubt, contributed to the manuscript and significantly improved it. Thank you.
General comments:
First, there are a number of grammatical issues (as well as some formatting issues, such as periods in the wrong place) and awkward sentences. There are too many to list, but these need to be addressed in order to provide clarity. The authors would do well to seek feedback from a native English speaker to help with these. In some cases, the ideas need to be better articulated, and this could be at least partially addressed with improvements to language usage.
A professional English scientific copy editor edited the entire revised manuscript. We believe this has improved the manuscript’s coherence, accuracy and structure.
Second, the objectives need to be more clearly stated; at some point it seems to the reader that the authors may have developed methods, though it is not clear whether this is in fact the case. If so, this needs to be integrated into the introduction as an objective, clearly stated in the methods, and needs to better articulated and discussed in the discussion/conclusions. What are the pre-existing methods, and what advantages can be brought by this method.
The reviewer is correct; this issue should be clarified.
The specific identification of F. oxysporum f. sp cepae was made using new primers designed in this work. As previously mentioned in the text (lines 157-159), “The FOC-SIX3 was designed using Primer3Plus software (http://primer3plus.com) on a region of the SIX3 coding sequence common to F. oxysporum f. sp cepae and Fusarium oxysporum f. sp. lycopersici.”
The following explanation was added to the text in the Materials and Methods section (lines 156-157): “The specific identification of F. oxysporum f. sp cepae was done using new primers designed in this work.”
The following sentence in the Results section was updated to emphasize this (lines 244-247): “The identification of the F. oxysporum isolates as f. sp. cepae relies on the highest sequence similarity scores of the studied genomic regions (Table 2, Figure 5A), and on the amplification of a product at the expected length when using the new specific FOC-SIX3 primers designed in this work (Table 1, Figure 5B1).”
The seedling pathogenicity assay and the onion bulbs inoculation assay are common inspection methods in plant pathology. These assays are widely used in various cases of host-pathogen interactions with some adaptions and variations. We made some adjustments to these methods and used them here to evaluate the isolates’ aggressiveness and the onion cultivars’ susceptibility. The following changes were made to the text (explanations were added and sentences were rewritten):
- Lines 335-338 – the following sentence was added: “The seedling pathogenicity assay and the onion bulb inoculation assay (that will be detailed in the following section) are common inspection methods in plant pathology. These assays are widely used in various cases of host-pathogen interactions with some adaptions and variations, as was done in this study.”
- Lines 338-339 – “The onion seedling assay was adjusted as an effective and rapid way to inspect the virulence capability of the Fusarium”
- Lines 389-391 – “This study identified different Fusarium species as the causal agent, and it adjusted tools to identify the fungi and assay isolate virulence, as well as enable the screening of onion germplasm for resistance.”
- Lines 440-441 – “The onion seedlings and bulbs pathogenicity assay that was adjusted and used in the current work ...”
Additionally, some comments are offered about fungicide screening, but this was not done in the current work, and the discussion on this topic is not very convincing simply because it was not properly articulated.
We agree. The following paragraph in the Discussion section (lines 440-447) was rewritten: “The onion seedlings and bulbs pathogenicity assay that was adjusted and used in the current work may be useful for breeding onion cultivars resistance and for screening fungicides, as was done in other crop fungal diseases (see, for example [31]). Also, this assay may enable us to examine other treatments and to study the penetration and establishment stages of the pathogenesis. A scientific program dedicated to developing disease control cannot be established exclusively on field experiments during the growing season. This is due to the considerable investment involved in such experiments, the long period until results are received, and the fluctuations in environmental conditions that cause inconsistency in the results.”
Finally, the authors suggest that species differences may be related to cultivar difference; however, there are several other possibilities that have not been discussed. For example, the identification of different species on the two cultivars may be related to geographic considerations (soil conditions, environmental conditions), or a preference for different colors--anthocyanins and other pigments may influence plant-pathogen interactions as well as disease resistances. I suspect more work is needed to answer these questions, but the discussion seems incomplete in this context.
This is a good point that should be addressed. The following paragraph in the Discussion section was rewritten (lines 409-415): “It should be taken into consideration that several other factors may be involved in the two unique mycoflora separations. For example, the identification of different species in the two cultivars may be related to geographic considerations (soil conditions, environmental conditions), or a preference for different colors – anthocyanins and other pigments may influence plant-pathogen interactions, as well as disease resistances. These interesting variations should be examined more thoroughly in future studies.”
Specific comments:
Line 23 – Fusarium-specific genes or primers?
Primers. The sentence was corrected to: “Final confirmation of the pathogens was performed with PCR amplification and sequencing using fungi-specific and Fusarium species-specific primers.” (lines 21-23)
Line 23 – field samples of what? soils? onions? onion crop debri?
Onion bulbs. The sentence was corrected to: “Four Fusarium spp. isolates were identified in onion bulbs samples collected from the contaminated field…” (lines 23-24)
Line 48, 461 – Fusarium species are widely distributed in what soil and associated with what plants?
The sentence was rewritten to clarify this and now reads: “Species of the fungus genus Fusarium are widely distributed in soils in all climatic zones around the globe, are associated with a vast diversity of plants, and are the cause of severe plant diseases in crops [2].” (lines 49-51)
Line 56 – F. oxysporum most damaging pathogen or plant pathogen?
Corrected to plant pathogen.
Line 65 - personal communications with whom?
This is a typo made by the Endnote bibliographic program. It was removed.
Line 85 - is it onion strains? or should it be cultivars/varieties?
Corrected to onion cultivars.
Line 113 - recommend to say ‘pure colony’ rather than clean.
Corrected to ‘pure colony’ as advised.
Line 122 - please clarify what is meant by single source.
The following explanation was added to the text (lines 125-127): “These colonies served as a single pure genetic source (a colony that has a common genetic origin - all cells evolved from one mother cell) for further research.”
Line 127 - obtained instead of received.
Corrected as advised.
Line 127 - please specify media.
We couldn’t find the word “media” in the specified location (line 127). We added an explanation to this word in the Results section where it first appeared. The sentence was corrected as advised and is now written: “The isolates grew well on solid and liquid-rich media.” (lines 274-275)
Line 134 – cover slip or glass slide?
Glass slide, as already written.
Microscopy - it is not clear whether the 10 uL drops of conidia in PDB or DDW was used to inoculate nutrient agar. This paragraph should be revisited for clarification on the steps used.
We modified the sentence to clarify that the suspension of mycelial mats in 10 ml PDB or DDW was used for the microscopy observation: “Mycelial mats or conidia were scraped off the plate, and a small amount was suspended in 10 µl PDB or DDW and placed on sterile glass slides for the microscopy observation.” (lines 136-137)
In the Fungal Slide Culture Technique, a block of inoculated nutrient agar (taken from 2-4-day-old agar that was used in the PDA colony). The sentence now reads: “A block of inoculated nutrient agar (taken from 2-4-day-old PDA colony) sandwiched between two sterile cover glass slides was placed in a plastic petri dish containing water agar.” (lines 139-141)
Line 142 – which TEF gene (i.e. 1- alpha, or another kind?); are the primers specific to Fusarium oxysporum species? or specific to different Fusarium species?
Corrected to “the Fusarium translation elongation factor - 1 alpha (TEF1) gene.”
As already mentioned in the text, the TEF1 primers are Fusarium-specific (i.e., specific to different Fusarium species).
Line 160 – was electrophoresis for size determination or for gel purification prior to sequencing? or both?
Both. The sentence was corrected and now reads: “For size determination and gel purification prior sequencing, the PCR products were separated on 1.5% agarose gel electrophoresis (Lonza, Rockland, USA) and sequenced (Hy Labs, Rehovot, Israel).” (lines 171-173)
Table 1 – F- is on separate line from primer for one primer; typo - multiplay lengths?
The separated line in each primer differentiates between the forward and the reverse primers used in each PCR reaction.
Riverside (Orlando) cultivar has been described as white or yellow in different places in the text.
Indeed, it can be described by both colors. We corrected all the descriptions to white.
Section 2.5 – please clarify what are the 6 replications? petri dishes with 15 seeds each?
Yes. The following sentence was rewritten: “The test was repeated twice with six replicates (petri dishes with 15 seeds each).” (lines 187-188)
Section 2.5 – Is this a seed assay or seedling assay?
It is a seedling assay. The title of this section was corrected to: “Seedling pathogenicity assay.” (line 185)
Line 191 – Whatman spelling, here and elsewhere in the manuscript
Corrected.
Section 2.6 – Onion bulb inoculation assay? or pathogenicity assay?
Corrected to: “Onion bulb pathogenicity assay.”
Line 199 – bulbs were inoculated in a moisture bag-- please specify how they were inoculated, with how much of what? Also, what is a moisture bag? in a moist bag?
This sentence in line 199 was modified as advised by Reviewer 1 and no longer includes mention of the moisture bag. A detailed description of the moist conditions is written in lines 207-209: “Each bulb was kept individually in a closed sterilized plastic bag to maintain a moist environment and prevent unwanted contamination in a temperature-controlled incubator in the dark at 28±1°C.”
Line 201 – biological hood? or biohazard hood? Was the entire assay carried out in the hood, or was set up in the hood?
The sentence was corrected to clarify this: “The entire assay was performed in six repetitions per isolate in a sterile environment within a biological hood.” (lines 200-201)
Lines 208-209 – how were disease symptoms evaluated?
The sentence was rephrased to clarify this point: “After two weeks of incubation, external and internal disease symptoms were photographed and evaluated qualitatively, and the fungus from each infected onion was re-isolated on PDA and identified to satisfy Koch’s postulates.” (lines 209-211)
Line 245 – isolates B1? or isolate B1?
Corrected to “isolate B1” (line 277)
Line 275 -– F. proliferatum is a fast-growing species--is this commonly known, or was this an observation made here. If the former, then a reference is needed, if the latter, perhaps the authors mean fastest growing among the species evaluated. Further to this, if this observation was made in a single isolate, then this could simply be a fast-growing isolate and not necessarily an F. proliferatum trait. Some clarity is needed in the writing.
The reviewer is correct. The sentence was rephrased as follows: “F. proliferatum (isolate B1) was the fastest growing among the isolates evaluated in this study, reaching a growth rate of 1.40 cm/day.” (lines 277-278)
Line 248 – colony color is not ideally evaluated in PDA as there can be variation among batches of media.
We agree, but all colonies that were photographed were grown on the same batch (bottle) of media. Moreover, because this medium is one of the most commonly used media for fungal growth and isolation, it is vital to present the colonies’ phenotype when grown on that particular nutrition agar.
The following explanation was added to Figure 4 legend: “Colonies were grown on potato dextrose agar (PDA; Difco Laboratories, Detroit, MI, USA) medium (prepared from the same batch) for seven days at 28±1°C in the dark.” (lines 287-288)
Line 249 – cultured? or cultures?
Corrected to “cultures.”
Table 3 – Fusarium spp. would be better description of the identification column.
Corrected as advised.
Please check spelling in table, and correct typos: Colony Growth; Hyphal Width.
Corrected as advised.
Please check that growth rates described in the text match those presented in the tables. There seems to be a disconnect in some of the values.
Indeed, some of the growth rates described in the text needed correction. We corrected the typos and verified that everything is accurate.
Figure 4 – description missing for ‘D.’
D in Figure 4 is a magnification of the conidiophores. We added this information to the figure legend.
Section 3.2 title – identification of what?
Corrected to: “Morphological and microscopic characterization and identification of the Fusarium isolates.”
Section 3.3 – title is awkward.
Corrected to “Molecular identification of the Fusarium isolates.”
Line 288 – Gene Bank or GenBank?
Corrected to “GeneBank”
Lines 242-243 – not clear what amplified sequence.
The word “amplified” was omitted from the sentence. The sentence now reads: “All four species examined presented high homology (99.4%-100% similarity) to previously described Fusarium spp. sequences in the GeneBank (Table 2).” (lines 243-244)
Line 294 – Sentence is not clear and should be rephrased.
We agree and think that this sentence does not fit in this section and can be omitted (the data are presented in detail in subsequent sections). Thus, the sentence was omitted from the text.
Once the acronyms have been defined at first use, do not need to spell them out again (i.e. ITS, TEF), this goes for figures too.
Corrected as advised throughout the text.
Figure 5 – there is a redundant sentence regarding primers.
Right, the redundant sentence was removed.
Figure 5 – this would be better as a supplementary file.
This is also a good option, but since the other three reviewers didn’t recommend it, we will leave that decision to the editor.
First sentence in Section 3.4 is awkward and should be rephrased.
The sentence was shortened and rephrased to: “Sequences alignment of the six Fusarium isolates, the TEF1 gene, revealed a highly conserved nucleotide array (Figure 6).”
Line 309 – Figure 6A or B?
Corrected to “Figure 6B.”
Line 311 – we’re or were?
Corrected to “were.”
Lines 316, 401 – two white and red onion cultivars - two of each? one of each? not clear?
The sentence was rephrased to clarify this: “Together with this, the two groups of the Fusarium species were also isolated from different onion cultivars (white and red); thus, the onion cultivar may be the cause of the two unique mycoflora.” (lines 322-324)
Lines 316-317 – are the authors suggesting that the cultivar may influence the microflora? Or that the type of onion would do this? Please clarify.
Indeed, as written in our answer above to your general comments, we suggest that the cultivar may influence the microflora. However, other factors may be involved, as you mentioned. Thus, we elaborate on this issue in the Discussion: (lines 410-415): “It should be taken into consideration that several other factors may be involved in the two unique mycoflora separations. For example, the identification of different species in the two cultivars may be related to geographic considerations (soil conditions, environmental conditions), or a preference for different colors – anthocyanins and other pigments may influence plant-pathogen interactions, as well as disease resistances. These interesting variations should be examined more thoroughly in future studies.”
Line 328 – onion seedling assay developed--where was this developed? If the method developed here that was not made clear.
As explained in our response to your general comments, we made some adjustments to the seedling assay method already used in other cases. We elaborated on this and made the necessary changes to the text to clarify this issue.
Line 329-331 – This phrase is awkward. Suggest reworking this to better express ideas. Furthermore, this seems more of a discussion point than a result.
Indeed, this phrase was removed, and the entire paragraph was reworked and edited according to your and Reviewer 1’s suggestions. The paragraph now reads: “Fusarium spp. can infect onion plants in many different ways, and the visual symptoms can be observed on plant leaves, roots, basal stem plate, and the bulb scales of small seedlings, mature plants and dormant bulbs [25,26]. The seedling pathogenicity assay and the onion bulb inoculation assay (that will be detailed in the following section) are common inspection methods in plant pathology. These assays are widely used in various cases of host-pathogen interactions with some adaptions and variations, as was done in this study. The onion seedling assay was adjusted as an effective and rapid way to inspect the virulence capability of the Fusarium isolates.” (lines 333-339)
Figure 8 – Any comment on the higher germination rate of treatments in some cases
Indeed, this information is missing in the description. The paragraph was corrected as advised: “The presence of F. proliferatum (isolate B1) or F. acutatum (B7, B5) significantly suppressed, and the presence of F. oxysporum f. sp. cepae (B8/B14) or F. anthophilium (B16) significantly increased, the seed germination of onion seeds (Riverside cultivars) under controlled conditions in petri dishes after nine days compared to the control (P < 0.05, Figure 8). No difference in the damping-off effect was found between F. oxysporum f. sp. cepae (B8/B14) or F. anthophilium (B16) and the control. The Noam cv. seeds’ germination was not affected by the inoculation with the Fusarium isolates B1, B5, B7 and B14. However, the germination was significantly enhanced by the isolates B8 and B16 (P < 0.05).” (lines 342-348)
Line 341, 370 – epicotyl or hypocotyl.
Hypocotyl. The term was corrected.
Figure 7 – if the seeds are not evenly spaced, could they have an interaction with each other, an increase or decrease in apparent pathogenicity? Mycelial growth on one can spread to another.
This is possible, but we think that if such an influence exists, it is insignificant. During the duration of the seedling assay, the pathogen hypha are spread all over the plate and reached all of the seeds.
Line 386 – if this study developed tools, it is not clear in the manuscript what those tools are.
We addressed this topic widely in our response to the general comments above.
Please check that the term isolate and species are used in the right places. For example, Line 396--should this be the identity of four species?
Corrected to: “the identity of four species.” We double-checked and corrected this throughout the text.
Noam is more resistant than Riverside. Is there any knowledge as to whether there are inherent differences in white vs red, or is this something that has not yet been explored?
As far as we know, this is something that has not yet been explored.
Line 407 states that all Fusarium isolates evaluated “significantly impact seed germination rate.” In Figure 8 some isolates increased germination rate. How can this be explained?
That is a good question. We don’t know if the increase in germination percentages in the presence of some of the isolates (B8, B14 and B16) is a direct consequence of the fungal-host interactions or the involvement of other factors.
It should be taken into consideration that a germinating seed was defined as a seed in which the seed coat was broken by the radicle. As clearly seen in Figure 7, there are great variations among the seeds of the same treatment. Some have a very small radicle with no hypocotyl. In contrast, others have well-developed radicle and hypocotyl. Thus, this measure may not reflect well the emergence inhibition in the plates.
However, the other measures – the hypocotyl elongation and the seedling biomass – are both negatively affected by all Fusarium species inspected here.
The following explanation was added to the text: “The increase in germination percentages in the presence of some of the isolates (B8, B14 and B16) may not reflect well the emergence inhibition in the plates. As seen clearly in Figure 7, there are variations among the seeds of the same treatment. Some have a very small radicle with no hypocotyl. In contrast, others have a well-developed radicle and hypocotyl. Thus, the hypocotyl elongation and fresh biomass better represent the influence of the pathogen under these conditions.” (lines 417-421)
Line 409 – “in over cases” is there a number missing here?
A reference number was added to the sentence: “… as demonstrated in other cases [29].”
Line 421 – is there an extra or missing word? “for” what?
Indeed, the term “for” is a typo and was removed from the text.
Line 427 – useful for breeding - as a selection tool? a screening tool? please clarify
Useful for breeding onion cultivars resistance and for screening fungicides. The following explanation was added: “The onion seedlings and bulbs pathogenicity assay that was adjusted and used in the current work may be useful for breeding onion cultivars resistance and for screening fungicides, as was done in other crop fungal diseases (see, for example [31]).” (lines 440-442)
Line 435 – prognostication about what?
The sentence was corrected to clarify this: “It is important to combine those two assays in order to achieve a more accurate prognostication about the aggressiveness of the pathogens involved in the onion basal rot disease.” (lines 450-451)
Lines 437-439 – discussion on treatments and how pot assays are an important first step, but the authors failed to first describe how these treatments could be applied in a pot assay, or if others have done this for onion.
We are currently working on a pot assay for screening fungicides to control the onion basal rot disease. We believe this should be the focus of a separate future paper. The following sentence was added to the text: “A pot assay for screening fungicides to control the onion basal rot disease should be the focus of a subsequent study.” (lines 455-457)
Line 455” as will be detailed below” is a strange way to end a discussion.
Correct, the phrase “as will be detailed below” was omitted from the text.
The discussion should be comprehensive in its own right, whereas the conclusion is more to summarize results and discussion.
The Conclusion section was shortened, as suggested, and now focuses on summarizing the Results and Discussion, as indicated.
Line 459 – “Rot in these plants”, better to be specific here.
The Conclusion section was modified according to Reviewer 1’s suggestion. This sentence was removed.
Reviewer 3 Report
Kalman et al. present an article on Fusarium species in onion in Israel. The article is well written, the isolation of strains and subsequent analyses were well conducted. I don’t have major comments on the science presented, except that in figure 8, in addition to testing seeds from red and white onions, the authors could also do the statistics to describe whether each isolates are significantly different from the control.
My other concerns are the low number of samples presented, six samples and the very strong focus of the authors on Israel, their home country. This is understandable. However, this article is going to be read [hopefully] by people from a lot of different countries. It would be important to include and develop on the state of research and threat caused by Fusarium on onion and other plant species in different countries and continents to give these results a more global dimension. I’m especially lacking a discussion of why this research matters other than at a local scale for onion producers.
Minor issues:
Line 132: write in whole letters potato dextrose broth (PDB)
Figure 2 A: I’m struggling to grasp what the 3rd image describe
Author Response
Responses to Reviewer 3’s comments
We thank the reviewer for investing time and effort, which contributed to this manuscript. The helpful and important remarks and suggestions improved this scientific paper and made it more accurate, clear and focused.
General comments:
In Figure 8, in addition to testing seeds from red and white onions, the authors could also do the statistics to describe whether each isolate is significantly different from the control.
As described in the Materials and Methods and the Figure 8 legend, this is indeed what we did: disease symptoms were analyzed by one-way ANOVA followed by post-hoc multiple comparisons of Student's t-Test for each pair (between the treatments and between the treatments and the control). In Figure 8, different letters above the error bar indicate a significant difference (p < 0.05); the same letters indicate no significant difference.
My other concerns are the low number of samples presented, six samples, and the very strong focus of the authors on Israel, their home country. This is understandable. However, this article is going to be read [hopefully] by people from a lot of different countries. It would be important to include and develop on the state of research and threat caused by Fusarium on onion and other plant species in different countries and continents to give these results a more global dimension. I’m especially lacking a discussion of why this research matters other than at a local scale for onion producers.
We agree and think that these important aspects should be included in subsequent studies.
The following part was added as a closing paragraph of the discussion, as advised: “This study on the mycoflora that causes onion basal rot disease in northeastern Israel has implications beyond the local scale and may contribute to onion producers in other regions. Others can follow the methodology used here. It may help to distinguish between the Fusarium species involved, which can be challenging due to the rapid changes of this particular pathogen genus. It would be important to continually widen the research on the threat caused by Fusarium on onion to other plant species in different countries and continents. These essential aspects should be researched in subsequent studies.” (lines 474-480)
Minor issues:
Line 137: write in whole letters potato dextrose broth (PDB).
Corrected as advised.
Figure 2A: I’m struggling to grasp what the 3rd image describes.
Right, this indeed should be clarified. The following explanation was added to the Figure 2 legend: “The two pictures on the left are a closeup of the symptoms, and the 3rd image is a wide view of the field from which these bulbs were collected. The density of the onion bulbs in the area may facilitate the disease spread.” (lines 232-235)
Reviewer 4 Report
The paper entitled “Isolation and Identification of Fusarium spp., the Causal Agents of Onion (Allium cepa) Basal Rot in Northeastern Israel” in an extensive work about an important plant disease.
It is well written, and I do to notice any significant flaw. The M&M are appropriate and the research is rigorous. Findings are consistent with objective and M&M.
However, this kind of research is generally published under “disease note” format. In fact, this disease is well known, and the novelty of this research seems to be limited to the spread in a yet-unreported area. All Fusarium species are, more or less, associated to onion bulb rot.
So, I suggest to change the format of the paper in a short communication or turn it in an epidemiologic study, investigating the spread of species in the area and/or during time.
Author Response
Responses to Reviewer 4’s comments
We thank the reviewer for investing time and effort, which contributed to this manuscript. Your contribution is greatly appreciated.
This kind of research is generally published under “disease note” format. In fact, this disease is well known, and the novelty of this research seems to be limited to the spread in a yet-unreported area. All Fusarium species are, more or less, associated to onion bulb rot. So, I suggest to change the format of the paper in a short communication or turn it in an epidemiologic study, investigating the spread of species in the area and/or during time.
This is indeed an option to consider. However, we think that the manuscript in its current form is more suitable to present, and to include all the results and discuss their meaning.
The study presented advances our understanding of the nature of this plant disease, the causative agents involved and their spread, and the damages they cause, and offers a new way of monitoring them.
The three other reviewers did not suggest changing the manuscript format. As written by Reviewer 1, the study increases the knowledge of the pathogenicity of Fusarium spp. fungi and may help to develop protection against Fusarium-caused onion basal rot.
Round 2
Reviewer 4 Report
I still think that a disease note format is more appropriate for this kind of report. However, I confirm the high quality of the paper, even more accurate in this revision. Thus, I suggest the pubblication of the paper in the present form.